# Stability and asynchrony of local communities but less so diversity increase regional stability of Inner Mongolian grassland

**Yonghui Wang[1], Shaopeng Wang[2], Liqing Zhao[1], Cunzhu Liang[1], Bailing Miao[1], Qing Zhang[1], Xiaxia Niu[1], Wenhong Ma[1]\*, Bernhard Schmid[3]\***

[1]Ministry of Education Key Laboratory of Ecology and Resource Use of the Mongolian Plateau & Inner Mongolia Key Laboratory of Grassland Ecology, School of Ecology and Environment, Inner Mongolia University, Hohhot, China; [2]Institute of Ecology, College of Urban and Environmental Sciences, and Key Laboratory for Earth Surface Processes of the Ministry of Education, Peking University, Beijing, China; [3]Department of Geography, Remote Sensing Laboratories, University of Zürich, Zürich, Switzerland

**\*For correspondence:**
whma@imu.edu.cn (WM);
bernhard.schmid@uzh.ch (BS)

**Competing interest:** The authors declare that no competing interests exist.

**Abstract** Extending knowledge on ecosystem stability to larger spatial scales is urgently needed because present local-scale studies are generally ineffective in guiding management and conservation decisions of an entire region with diverse plant communities. We investigated stability of plant productivity across spatial scales and hierarchical levels of organization and analyzed impacts of dominant species, species diversity, and climatic factors using a multisite survey of Inner Mongolian grassland. We found that regional stability across distant local communities was related to stability and asynchrony of local communities. Using only dominant instead of all-species dynamics explained regional stability almost equally well. The diversity of all or only dominant species had comparatively weak effects on stability and synchrony, whereas a lower mean and higher variation of precipitation destabilized regional and local communities by reducing population stability and synchronizing species dynamics. We demonstrate that, for semi-arid temperate grassland with highly uneven species abundances, the stability of regional communities is increased by stability and asynchrony of local communities and these are more affected by climate rather than species diversity. Reduced amounts and increased variation of precipitation in the future may compromise the sustainable provision of ecosystem services to human well-being in this region.

## Editor's evaluation

Wang et al. adapt a new statistical framework on a multisite multiyear database to investigate the effects of environmental variables on the temporal stability of plant communities and biomass productivity in Chinese grassland. The authors show that the temporal stability of the region is due to spatial asynchrony of community dynamics. This article will be a landmark in the field as it sets a new methodological framework to study the impacts of global warming in ecosystems and conservation biologists and politicians to design regional policies for land management.

## Introduction

The ability of ecosystems to stably provide biological products and services such as biomass production for human well-being (*Isbell et al., 2015*; *Tilman et al., 2014*, *Tilman et al., 2006*) is being threatened by species loss (*Cardinale et al., 2012*; *Harrison et al., 2015*; *Isbell et al., 2017*; *Isbell*

*et al., 2015*; *Newbold et al., 2015*; *Tilman et al., 2014*) and climate change (*Hautier et al., 2015*; *Hautier et al., 2014*; *Ma et al., 2017*; *Xu et al., 2015*). Policymakers seek guidance to make management and conservation decisions at high levels of ecological organization, for example, for an entire region with diverse plant communities (*Cardinale et al., 2012*; *Isbell et al., 2017*; *Parker et al., 2019*; *Wang et al., 2019*), here referred to as a regional community. However, previous theoretical, experimental, and observational studies on ecosystem stability have mostly been conducted at local scales with homogenous environmental conditions (*Hautier et al., 2015*; *Hautier et al., 2014*; *Hector et al., 2010*; *Isbell et al., 2015*; *Ma et al., 2017*; *Tilman et al., 2006*; *Wang et al., 2020*). Patterns of ecosystem stability discovered in local communities may not directly scale up to a system of spatially separate communities (*Lamy et al., 2019*; *McGranahan et al., 2016*; *Wang et al., 2019*; *Wang and Loreau, 2016*; *Wang and Loreau, 2014*; *Wilcox et al., 2017*; *Zhang et al., 2019*). Thus, there is an urgent need to understand stability and the factors maintaining it at spatial scales covering larger areas (*Gonzalez et al., 2020*; *Isbell et al., 2017*; *Wang et al., 2019*).

Recent theoretical work facilitates investigations of ecosystem stability at larger spatial scales (measured by the ratio of temporal mean to standard deviation of ecosystem productivity over time) by relating it to its hierarchical components along two alternative pathways I or II (*Wang et al., 2019*; *Box 1*). Along pathway I, local asynchronous species dynamics stabilize local communities (step A, species insurance effect of local species asynchrony; *Yachi and Loreau, 1999*) and asynchronous community dynamics among distant localities stabilize regional communities (step B, spatial insurance effect of regional community asynchrony; *Wang and Loreau, 2016*; *Wang and Loreau, 2014*). Along pathway II, asynchronous population dynamics within species among distant localities stabilize regional populations (step A, spatial insurance effect of regional population asynchrony; *Wang and Loreau, 2016*; *Wang and Loreau, 2014*) and regional asynchronous species dynamics stabilize regional communities (step B, species insurance effect of regional species asynchrony; *Wang et al., 2019*).

Species diversity has been hypothesized to stabilize ecosystems at different ecological hierarchies because species-rich communities are more likely to include species that have different responses to environmental variation across time and space, producing stable communities via species asynchrony (*Thibaut and Connolly, 2013*; *Tilman et al., 2014*; *Wang and Loreau, 2016*; *Wang and Loreau, 2014*; *Wang et al., 2020*). In natural ecosystems, the role of species diversity in affecting stability across different ecological hierarchies is still unclear. Theoretical and experimental studies propose stabilizing effects of (alpha) diversity within local communities (*Hautier et al., 2015*; *Hautier et al., 2014*; *Hector et al., 2010*; *Tilman et al., 2014*; *Tilman et al., 2006*). However, these studies usually consider systems in which species abundance distributions are relatively even, at least at the beginning of newly assembled communities in biodiversity experiments (*Hector et al., 2010*; *Tilman et al., 2006*). Natural communities are often characterized by highly uneven abundance distributions and dominated by the dynamics of a few abundant species (*Thibaut and Connolly, 2013*; *Wang et al., 2019*), even with exceptions that sometimes low-abundance species can disproportionately contribute to certain ecosystem functions (*Dee et al., 2019*; *Leitão et al., 2016*; *Schmid et al., 2022*). With highly uneven abundance distributions, the predicted local-scale diversity–stability relationship may be weak and it may be sufficient to focus on the population dynamics of dominant species (see Appendix 1 for dominant-species measure) instead of all species (*Wang et al., 2020*; *Xu et al., 2015*; *Yang et al., 2012*). Furthermore, theoretical studies also propose that the heterogeneity in species compositions between distant local communities (beta diversity) can increase asynchronous dynamics among them, resulting in stabilized regional communities (*Wang et al., 2019*; *Wang and Loreau, 2016*). Currently, empirical evidence for such an effect is mixed as it was detected in some (e.g., *Hautier et al., 2020*; *Liang et al., 2021*; *Qiao et al., 2022*; *Wang et al., 2019*) but not in other studies (e.g., *Wilcox et al., 2017*; *Yang et al., 2022*; *Zhang et al., 2019*). These studies looked at rather small spatial scales with potentially low beta diversity or even the same dominant species occurring among local communities, making it difficult to detect a stabilizing effect at the regional scale where management decisions are taken. Because different species may be dominant in distant local communities, asynchrony among these local communities may contribute to regional community stability (*Wang et al., 2019*; *Wang and Loreau, 2016*; *Wang and Loreau, 2014*; *Isbell et al., 2018*).

To investigate the temporal stability of aboveground biomass production ('productivity' for short) at larger spatial scales, we established a region-scale observation network in Inner Mongolian grassland

## Box 1. Theory and glossary.

Theoretically, the local population coefficient of variation (CV, i.e., the ratio of mean to standard deviation or the inverse of stability) can be stepwise upscaled to the regional community CV via either the local community CV (pathway I) or the regional population CV (pathway II) (**Box 1—figure 1**). In each step, synchrony (inverse of asynchrony) measures the proportion of CV upscaled from the lower to the higher hierarchical level. Specifically, synchrony takes value between 0 (perfectly asynchronous) and 1 (perfectly synchronous) and the CV at the higher level is the product of synchrony and the CV at the lower level (**Loreau and de Mazancourt, 2008**; **Thibaut and Connolly, 2013**; **Wang et al., 2019**). Along pathway I, the local population CV first upscales with local species synchrony to the local community CV (step A); then, the local community CV upscales with regional community synchrony to the regional community CV (step B). Along pathway II, the local population CV first upscales with regional population synchrony to the regional population CV (step A); then the regional population CV upscales with regional species synchrony to the regional community CV (step B).

In this study, we use superscripts $P$ and $C$ to designate hierarchical components at population and community levels, respectively, and superscripts $L$ and $R$ to designate local and regional scales, respectively. We use superscript $P{\to}C$ and $L{\to}R$ to designate scaling up from populations to communities and from local to regional measures, respectively. All measures were estimated with all species or only dominant species, the latter designated with subscript $d$.

*Temporal CV and synchrony*

**Local population CV ($CV^{P,L}$)**: Defined as the weighted-average local population CV of plant aboveground biomass across species and local communities. Hypothesis: positively or negatively influenced by alpha diversity (**Thibaut and Connolly, 2013**; **Tilman et al., 2006**).

**Local species synchrony ($\varphi^{P{\to}C,L}$)**: Defined as the weighted-average synchronous biomass dynamics among local populations within local communities. Hypothesis: negatively influenced by alpha diversity (**Loreau and de Mazancourt, 2008**; **Thibaut and Connolly, 2013**).

**Local community CV ($CV^{C,L} = CV^{P,L} \times \varphi^{P{\to}C,L}$)**: Defined as the weighted-average local community CV of biomass among local communities. Hypothesis: negatively influenced by alpha diversity (**Loreau and de Mazancourt, 2008**; **Thibaut and Connolly, 2013**; **Tilman et al., 2014**).

**Regional community synchrony ($\varphi^{C,L{\to}R}$)**: Defined as the weighted-average spatial synchronous biomass dynamics among local communities. Hypothesis: negatively influenced by beta diversity (**Wang et al., 2019**).

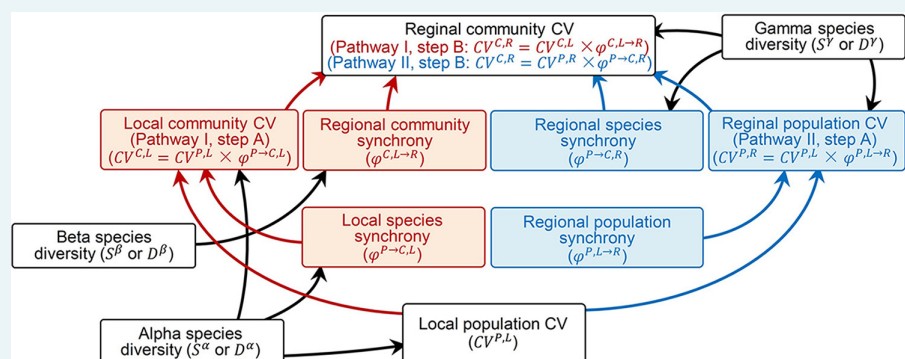

**Box 1—figure 1.** Upscaling local population coefficient of variation (CV) to regional community CV via local community CV (pathway I, red arrows on the left side) and regional population CV (pathway II, blue arrows on the right side), as well as theoretically proposed impacts of species diversity measures on them.

*continued on next page*

**Regional population synchrony ($\varphi^{P,L\to R}$)**: Defined as the weighted-average spatial synchronous biomass dynamics among local populations of the same species. Hypothesis: negatively influenced by spatial heterogeneity (**Wang et al., 2019**).

**Regional population CV ($CV^{P,R} = CV^{P,L} \times \varphi^{P,L\to R}$)**: Defined as the weighted-average regional population CV of biomass across species. Hypothesis: positively or negatively influenced by gamma diversity (**Wang et al., 2019**).

**Regional species synchrony ($\varphi^{P\to C,R}$)**: Synchronous biomass dynamics among regional populations of different species. Hypothesis: negatively influenced by gamma diversity (**Wang et al., 2019**).

**Regional community CV ($CV^{C,R}$)**: Defined as the CV of regional community biomass. Can be upscaled via aggregating local communities (pathway I, $CV^{C,R} = CV^{C,L} \times \varphi^{C,L\to R}$) or organizing regional populations (pathway II, $CV^{C,R} = CV^{P,R} \times \varphi^{P\to C,R}$). Hypothesis: negatively influenced by gamma diversity (**Wang et al., 2019**).

*Species diversity*
**Alpha species diversity**: Local species richness ($S^\alpha$) or effective species richness ($D^\alpha$).
**Beta species diversity**: Cross-locality dissimilarity of species richness ($S^\beta$) or effective species richness ($D^\beta$).
**Gamma species diversity**: Regional species richness ($S^\gamma$) or effective species richness ($D^\gamma$).

*Climatic variables*
**Mean annual precipitation (MAP)**: Cross-site averaged mean annual precipitation (or annual temperature, *MAT*).
**Local and regional precipitation CVs ($CV_P^L$ and $CV_P^R$)**: Measuring precipitation (or temperature, $CV_T^L$ and $CV_T^R$) variability with its local and regional CVs.
**Regional precipitation synchrony ($\varphi_P^{L\to R}$)**: Spatial synchronous dynamics of precipitation (or temperature, $\varphi_T^{L\to R}$).

in China across an area of >166,894 km² and monitored the yearly dynamics of productivity over five consecutive years (**Figure 1a**). The Inner Mongolian grassland represents a typical part of the Eurasian grassland biome and is crucial in providing biological products and services to human societies living there (**Fang et al., 2015**; **Kang et al., 2007**). In this region, plant community productivity and species richness and composition are driven by climatic factors, that is, temperature and precipitation (**Bai et al., 2004**; **Hu et al., 2018**; **Ma et al., 2010**; **Wang et al., 2020**; **Xu et al., 2015**). These have changed considerably during the past decades (**Huang et al., 2015**; **Piao et al., 2010**) with largely unknown ecological consequences, especially at large spatial scales.

To facilitate the large-scale stability investigation, we employed a simulated landscape method (**Hautier et al., 2018**; **van der Plas et al., 2016**) to construct large-scale, that is, regional communities consisting of two local communities (two observed sites) separated by 17–987 km (**Figure 1a**). Briefly, each regional community was constructed by randomly choosing two distant local communities without replacement (to ensure replicate regional communities were not sharing local communities; **Figure 1b**). We did not consider scenarios including more than two local communities in each regional community because the resulting small number of replicate regional communities would have prevented a statistical analysis (but see **Appendix 1—figure 3** for a three-local-community scenario). Based on the above framework, we investigated how asynchronous population dynamics among species, in particular dominant ones, at local and regional scale contributed to the stability of local and regional communities in the study region. We also tested how local and regional community dynamics were driven by species diversity or affected by climatic factors such as precipitation and its temporal variation. First, we analyzed stability variables with general linear models (GLMs) to identify important relationships. We then used this information together with theoretical considerations to construct path-analytic diagrams from structural equation models (SEMs) relating the regional community coefficient of variation (CVs; i.e., inverse of stability) of plant aboveground biomass to its hierarchical

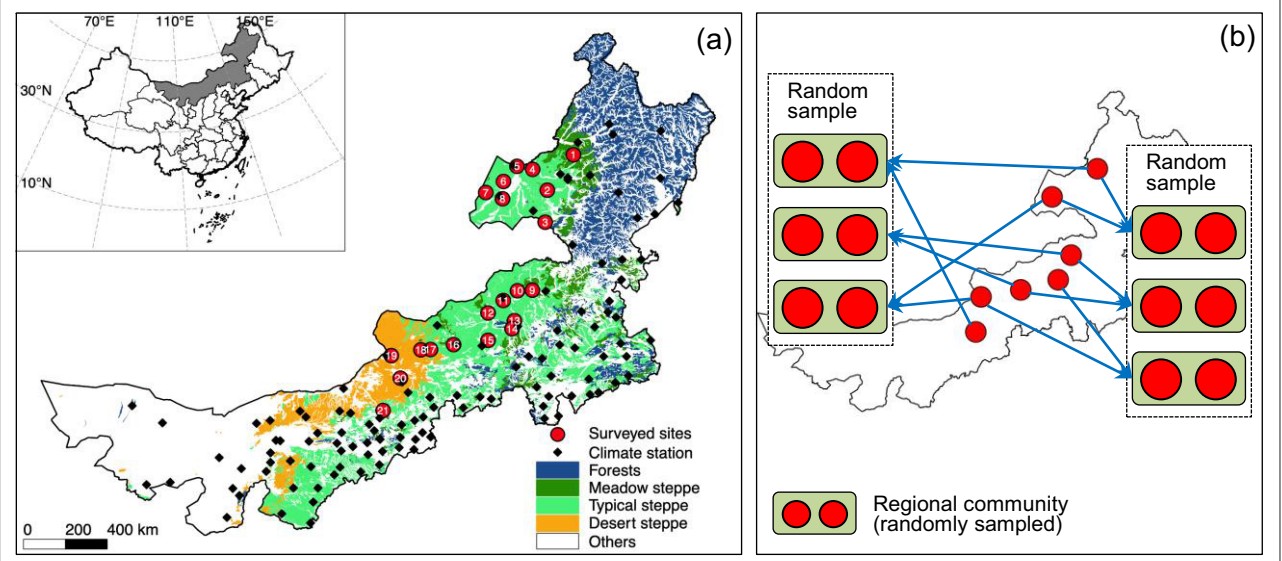

**Figure 1.** Geographic distribution of surveyed sites with site numbers (**a**) and a simplified case (seven-site) for illustrating construction of regional communities aggregating two local communities (**b**). In (**a**), red circles represent sites included in constructing regional communities. (**b**) shows a simplified case illustrating the construction of two sets of regional communities using random sampling without replacement to ensure the regional communities within each set do not share local communities.

The online version of this article includes the following figure supplement(s) for figure 1:

**Figure supplement 1.** Regional synchronies of temperature (**a**) and precipitation (**b**), all-species measure of regional community coefficient of variation (CV, inverse of stability, **c**) and all-species (**d–f**) and dominant-species (**g–i**) measures of regional community synchrony (inverse of asynchrony), regional population CV, and regional population synchrony in relation to distance.

components and all- or dominant-species diversity as well as climatic factors (*Thibaut and Connolly, 2013*; *Tilman et al., 2014*; *Wang et al., 2019*; *Wang and Loreau, 2016*; *Wang and Loreau, 2014*). As commonly done in such studies, all analyses were conducted using inverse values of stability and asynchrony, that is, CV and synchrony. We present an overview of the upscaling models and a glossary of terms in *Box 1*.

## Results
### Part I: Analysis using all species

We first analyzed variation in the regional community CV in relation to its hierarchical components including all species and in relation to all-species diversity indices as well as climatic factors. We found that the regional community CV was positively associated with the local community CV and regional community synchrony (step B of upscaling pathway I; *Figure 2a and b*, *Figure 3a*). The local community CV in turn was positively related to the local population CV and local species synchrony (step A in upscaling pathways I; *Figure 2e and f*, *Figure 3a*). Along the upscaling pathway I, the CVs decreased from 0.76 for the local population CV to 0.38 for the local community CV and further to 0.29 for the regional community CV (*Figure 3a*), as a result of a lower local species synchrony (mean = 0.49) compared with regional community synchrony (mean = 0.78; *Figure 3a*).

Alternatively, the regional community CV was positively related to the regional population CV and regional species synchrony (step B of upscaling pathway II; *Figure 2c and d*, *Figure 3a*). The regional population CV was in turn positively related to the local population CV but not to regional population synchrony (step B of upscaling pathway II; *Figure 2g and h*). However, the path from regional population synchrony to regional population CV was suggested by theory and therefore included in the SEM, where it became significant (*Figure 3a*). Along the upscaling pathway II, the CVs declined from 0.76 for the local population CV to 0.71 for the regional population CV and further to 0.29 for the regional community CV (*Figure 3a*), as a result of a higher regional population synchrony (mean = 0.94) compared with regional species synchrony (mean = 0.41; *Figure 3a*).

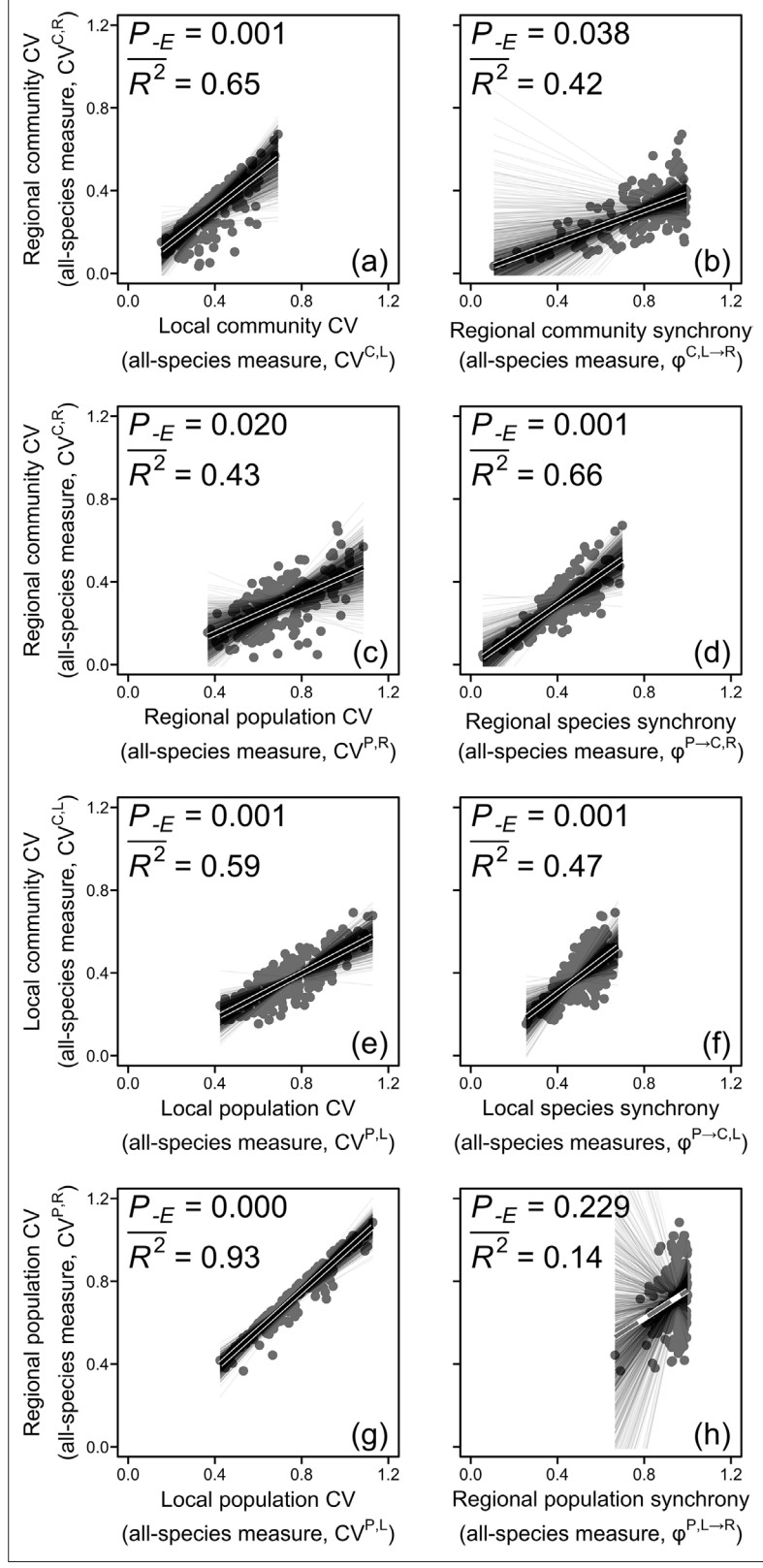

**Figure 2.** The regional community **(a–d)**, local community **(e–f)**, and regional population **(g–h)** coefficients of variation (CVs) in relation to their hierarchical components using all-species measures. Solid black lines represent significant (p<0.05) and marginally significant (p<0.10) relationships, and dashed gray lines represent nonsignificant (p>0.10) relationships (see 'Materials and methods' for details and **Box 1** for glossary). Thin grey lines represent

*Figure 2 continued on next page*

*Figure 2 continued*

relationships for 1000 sets of resampled regional communities (n=10 for each set). See **Figure 2—figure supplement 1** for results of using dominant-species measures. Dataset and code are available in Figshare at https://doi.org/10.6084/m9.figshare.20281902.

The online version of this article includes the following figure supplement(s) for figure 2:

**Figure supplement 1.** All-species estimate of regional community coefficient of variation (CV, inverse of stability) in relation to dominant-species estimates of local community CV (**a**), regional community synchrony (inverse of asynchrony, **b**), regional population CV (**c**), and synchrony (**d**) as well as dominant-species estimates of local community CV (**e, f**) and regional population CV (**g, h**) in relation to their hierarchical components.

We found that all-species diversity indices had relatively weak impacts on CVs and synchronies across ecological organization levels (*Figure 4a and c–e*; see 'Materials and methods' for calculating species diversity indices across scales). Although correlation (*Appendix 1—figure 2a*) and regression (*Figure 4b*) analyses showed that the regional population CV was positively related to gamma diversity, this was not supported by the final path analysis model (*Figure 3a*). However, the local population CV was positively and the local species synchrony negatively related to alpha diversity (*Figure 3a*, *Figure 4f and g*).

Local species synchrony increased with the local precipitation CV but no relations between the regional community CV or its other components and climatic variation were found (*Figure 3a*, *Figure 4h and i*).

## Part II: Analysis using only dominant species

Considering only dominant species in the hierarchical components was sufficient to explain a large amount of variation in regional community CV. For the upscaling pathway I, the regional community CV was positively related to local community CV and regional community synchrony (step B), with the explanatory power reduced to 0.71 from 0.98 for the analysis using all species (comparison of *Figure 3b* with *Figure 3a*). The local community CV in turn was positively related to the local population CV and to local species synchrony (step A; *Figure 3b*). For the upscaling pathway II, the regional community CV was positively related to the regional population CV and regional species synchrony (step B), with the explanatory power reduced to 0.69 from 0.98 for the analysis using all species. The regional population CV was positively related to the local population CV and regional population synchrony. Dominant species as a group explained more than half of the variance of CVs and synchronies estimated with all species (explanatory power, $\bar{R}^2$ ,>0.50, *Figure 3—figure supplement 1*), except for the regional population synchrony ($\bar{R}^2$ = 0.14, *Figure 3—figure supplement 1h*).

Similar to the analysis with all species, diversity indices of only dominant-species had relatively weak impacts on CVs and synchronies across organizational levels (*Figure 4—figure supplement 1a–g*). Although correlation analyses showed that the regional population CV was positively related to gamma diversity (*Appendix 1—figure 2b*), this was again not supported by the final path analysis model (*Figure 3b*). However, for dominant species, the local population CV was increased by alpha diversity and reduced by larger mean values of precipitation (*Figure 3b*, *Figure 4—figure supplement 1i*).

## Discussion

Based on a multiyear region-scale survey in Inner Mongolian grassland, we investigated stability (inverse of CV) and asynchrony (inverse of synchrony) across spatial scales and analyzed influences of species diversity, dominant species, and climatic factors on them. We found that the regional community stability was related to the stability and asynchronous dynamics of local communities. In addition, stability and asynchrony were — albeit weakly — impacted by species diversity. Compared with the dynamics of all species, the dynamics of only dominant species had also good predictive power, indicating that these species were important drivers of grassland stability in the region. Furthermore, decreasing mean and increasing interannual fluctuation of precipitation could, respectively, destabilize dominant species and synchronize population dynamics within local communities, impairing stability at the regional scale.

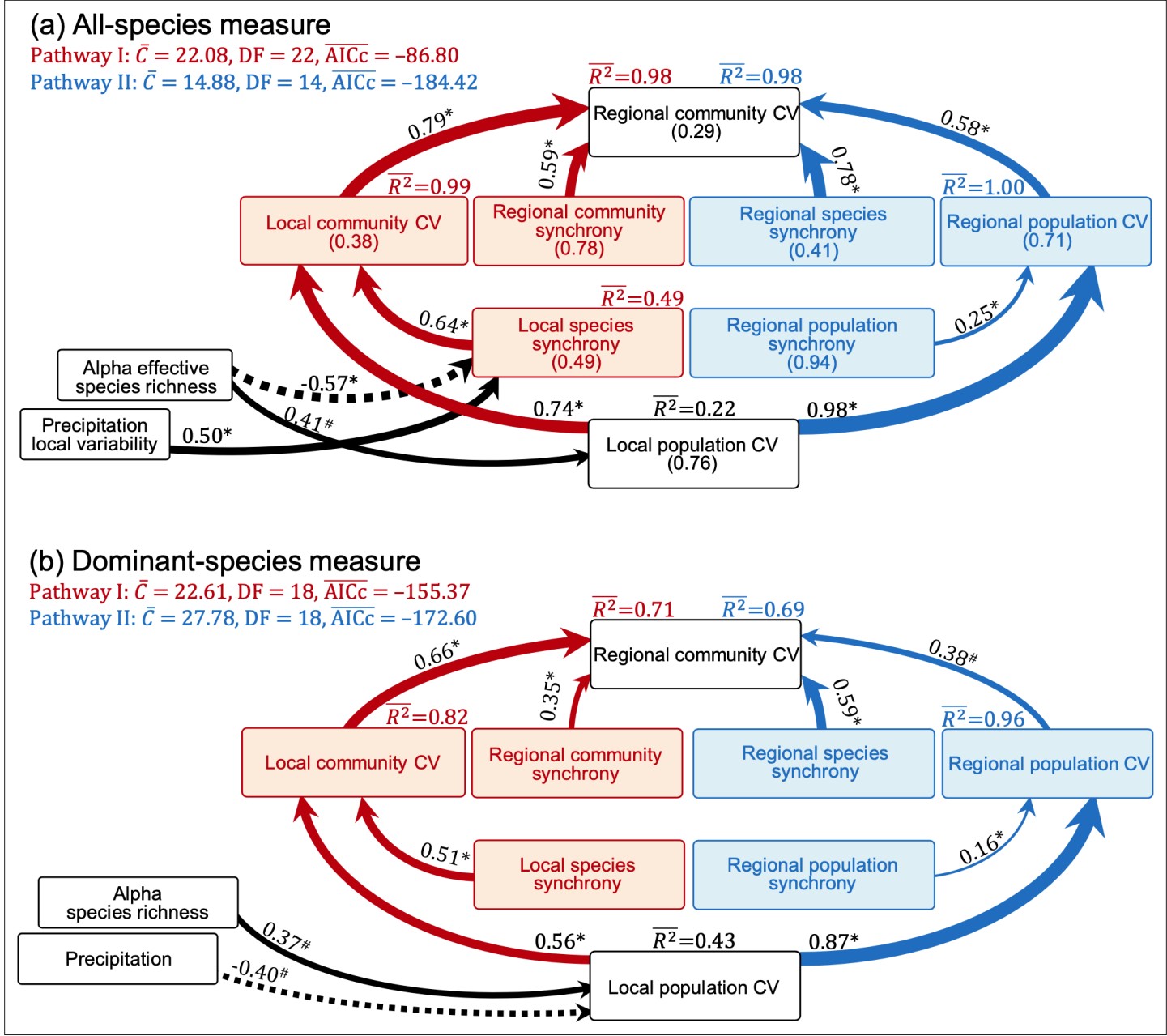

**Figure 3.** Path analysis models relating the regional community coefficient of variation (CV, all-species measure) to its hierarchical components and species diversity indices estimated with all species (**a**, the mean values of CVs and synchronies are in brackets) and only dominant species (**b**) as well as climatic factors. These diagrams combine different upscaling pathways (pathway I, left side with red arrows; pathway II, right side with blue arrows). Solid and dashed arrows, respectively, represent positive and negative paths, and numbers near arrows are standardized path coefficients. The significance level of each path is indicated by * when p<0.05 or # when p<0.10 (see 'Materials and methods' for details and *Box 1* for glossary). See *Figure 3—figure supplement 1* for relationships between all-species and dominant-species measures. Dataset and code are available in Figshare at https://doi.org/10.6084/m9.figshare.20281902.

The online version of this article includes the following figure supplement(s) for figure 3:

**Figure supplement 1.** All-species estimates (vertical axes) of coefficients of variation (CVs, inverse of stability) and synchronies (inverse of asynchrony) across hierarchical levels of ecological organization in relation to their dominant-species counterparts (horizontal axes) (**a–b**, regional community CV; **c**, regional community synchrony; **d**, regional species synchrony; **e**, local community CV; **f**, regional population CV; **g**, local species synchrony; **h**, regional population synchrony; **i**, local population CV).

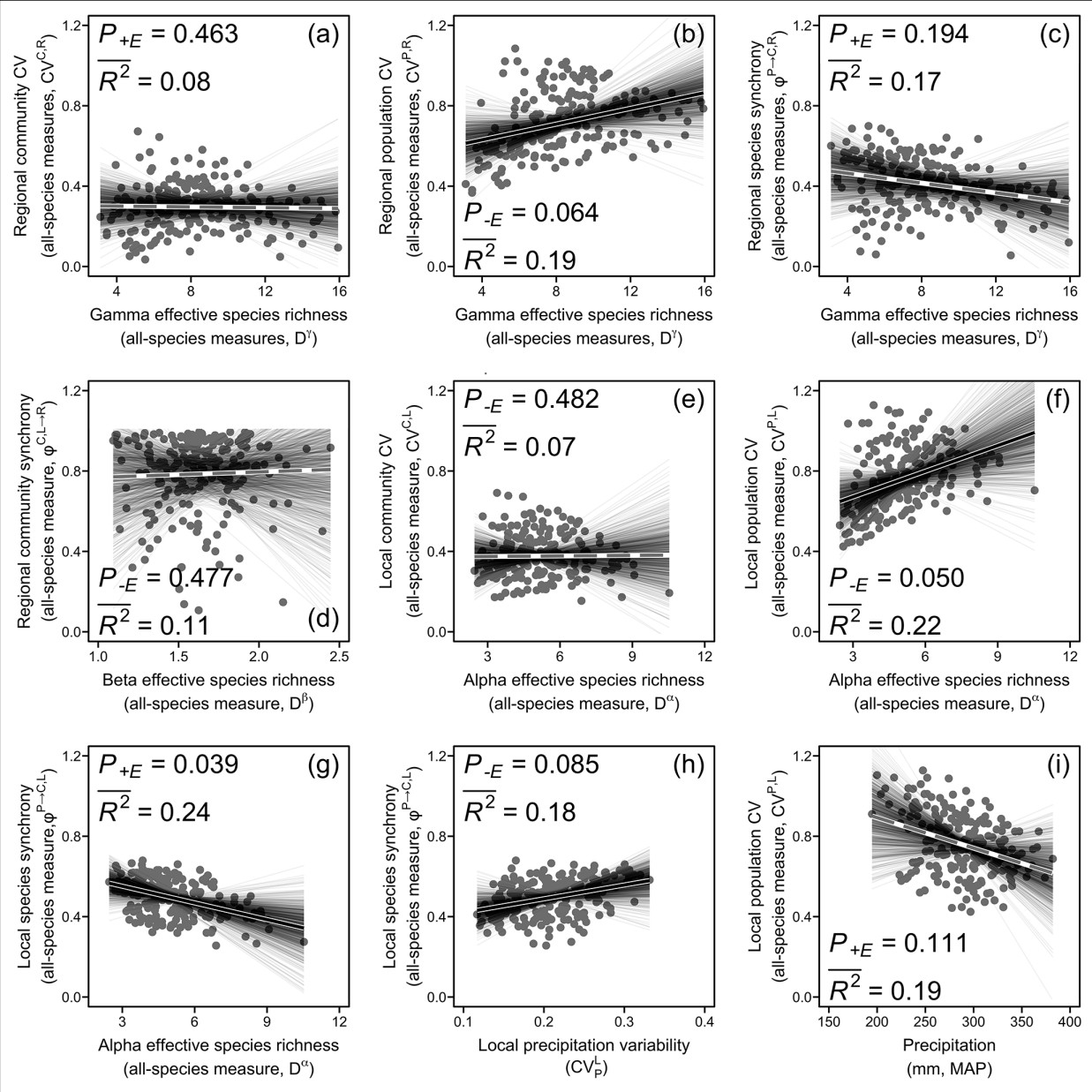

**Figure 4.** Regional community coefficient of variation (CV) (**a**), regional population CV (**b**), regional species synchrony (**c**), regional community synchrony (**d**), local community CV (**e**), local population CV (**f**), and local species synchrony (**g**) in relation to species diversity (effective species richness) as well as local species synchrony and local population CV, respectively, in relation to local precipitation variability (**h**) and precipitation (**i**) using all-species measures. Solid black lines represent significant (p<0.05) and marginally significant (p<0.10) relationships, and dashed gray lines represent nonsignificant (p>0.10) relationships (see 'Materials and methods' for details and **Box 1** for glossary). Thin grey lines represent relationships for 1000 sets of resampled regional communities (n=10 for each set). See **Figure 4—figure supplement 1** for results of using dominant-species measures. Dataset and code are available in Figshare at https://doi.org/10.6084/m9.figshare.20281902.

The online version of this article includes the following figure supplement(s) for figure 4:

**Figure supplement 1.** Regional community coefficient of variation (CV, inverse of stability) estimated with all species (**a**) and dominant-species estimates of regional population CV (**b**), regional species synchrony (**c**), regional community synchrony (**d**), local community CV (**e**), local population CV (**f**), and local species synchrony (**g**) in relation to species diversity (effective species richness) as well as dominant-species estimates of local species synchrony and local population CV in relation to local precipitation variability (**h**) and precipitation (**i**), respectively.

## Stability across ecological hierarchies

We investigated ecosystem stability across ecological hierarchies with two alternative upscaling pathways (*Wang et al., 2019*), and both of them showed gradually increasing stability from low to high organization levels due to species insurance effects and spatial insurance effects of populations and communities, caused by asynchronous dynamics among species and localities (*Figure 3a*). These patterns are consistent with recent studies carried out at single sites constructing multiple adjacent plots within meta-communities in grassland ecosystems (*Hautier et al., 2020*; *McGranahan et al., 2016*; *Wang et al., 2019*; *Wilcox et al., 2017*; *Zhang et al., 2019*) and at the regional scale in marine ecosystems (*Lamy et al., 2019*; *Thorson et al., 2018*), as well as recent theoretically proposed positive invariability–area relationships (*Isbell et al., 2018*; *Wang et al., 2017*). These results suggest that, at large spatial scales, spatial heterogeneity is important in maintaining stability; losing this heterogeneity (*Fahrig et al., 2011*; *Gámez-Virués et al., 2015*) can impair stability.

We found that species insurance effects (local species asynchrony; *Tilman et al., 2014*; *Yachi and Loreau, 1999*) were stronger in maintaining stability at the regional scale than the spatial insurance effects of distant populations and communities despite the large extent and thus expected spatial heterogeneity of our study region. This result is consistent with a recent investigation in marine plant communities (*Lamy et al., 2019*) but different from that in fish communities (*Thorson et al., 2018*). In our study, the region-wide synchronous variation in precipitation (mean = 0.86, ranged from 0.62 to 1.00) (*Figure 1—figure supplement 1b*) potentially decreased the spatial heterogeneity and increased the relative importance of the species insurance effect. The regulation of spatial insurance effects on the stability of fish communities at regional scale may result from their high mobility. Fish can move toward their preferred environmental conditions, causing more variable spatial population patterns than those found in plants, potentially strengthening the spatial insurance effects. In plant communities, the strong species insurance effect suggests that regional community stability to a large part reflects the stability of local communities, which have predominantly been considered in previous studies (*Ma et al., 2017*; *Tilman et al., 2006*; *Xu et al., 2015*; *Yang et al., 2012*).

## Influence of species diversity, dominant species, and precipitation on ecosystem stability

We only detected stabilizing impacts of species diversity at local but not at regional scale (*Figure 3*, *Figure 4f and g*). The observed negative species richness–local population stability relationship is in line with theoretical and experimental studies (*Lehman and Tilman, 2000*; *Tilman, 1999*; *Tilman et al., 2014*, *Tilman et al., 2006*), proposing that competition between coexisting species for resources in species-rich communities leads to low population stability. The observed positive species richness–local species asynchrony relationship is also expected by theory based on the higher probability of species-rich communities to contain species that are different in responding to environmental fluctuations (*Tilman et al., 2014*; *Yachi and Loreau, 1999*).

Previous studies reported mixed impacts of species diversity on stability and asynchrony at scales beyond local. Some studies found significant influences (e.g., *Hautier et al., 2020*; *Liang et al., 2021*; *Qiao et al., 2022*; *Wang et al., 2019*) and others found none (e.g., *Wilcox et al., 2017*; *Yang et al., 2022*; *Zhang et al., 2019*). It has been argued (*Hautier et al., 2020*) that investigations within a single site (*Zhang et al., 2019*) or multiple sites with nonstandardized experimental protocols (*Wilcox et al., 2017*) may mask stabilizing effects of species diversity at regional scale. However, even though this study used a multisite dataset across a large region with a standardized survey protocol, it still could not detect significant effects of species diversity at the regional scale (*Figure 3*). The highly uneven distribution of species abundances could in part have been responsible for this (*Wang et al., 2020*). Nevertheless, even when only dominant species were considered, we could still not find the expected relationship between species diversity and regional asynchrony and stability.

It is conceivable that in natural grassland ecosystems, which are often characterized by high unevenness (*Dee et al., 2019*; *Jiang et al., 2009*; *Smith and Knapp, 2003*), species have co-evolved over time in such a way as to maintain high stability at a level of species diversity established over a longer time span. This observation has been made in a long-term grassland biodiversity experiment, where a history of co-occurrence led to greater community stability in response to a flooding event (*van Moorsel et al., 2021*) and repeated exposure to drought led to increased species complementarity in response to drought (*Chen et al., 2022*). The absence of strong species diversity–stability

relationship at regional scales in observational studies is not the same as rapid biodiversity loss from an established level typically simulated in experiments. By extension, regional species loss in the study area due to, for example, land-use and climate change in the future may still pose a threat to regional grassland stability.

The strong influence of precipitation on the productivity of different species (*Zhang et al., 2017*) may have weakened insurance effects of species diversity in this study. Strong fluctuations in precipitation in dry grassland may force similar responses among different species, decreasing the dissimilarity and thus compensatory dynamics among species. This speculation is supported by the low local species asynchrony under high precipitation fluctuation (*Figure 3a*, *Figure 4h*). Furthermore, for the dominant-species measures, we also found decreased local population stability under low precipitation means (*Figure 3b*, *Figure 4—figure supplement 1i*), potentially due to the decreasing mean-to-standard deviation ratio caused by the dominant-species biomass production being more steeply related to precipitation amount than to its standard deviation (*Wang et al., 2020*). The study region has been experiencing a pronounced decrease in precipitation and an increase in its variability during the past decades (*Huang et al., 2015*; *Piao et al., 2010*; *Tao et al., 2015*). Our results indicate that these changes in precipitation regimes may present a key threat to the sustainable provision of biological products and services to human well-being in the region.

## Materials and methods
### Study region and plant community survey

The Inner Mongolian temperate grassland has a continental monsoon climate with a short and cool growing season (from May to October, averaged temperature 12.9–18.4°C across sites during the studied period from 2012 to 2016), concentrating ~90% of the annual precipitation (averaged precipitation 186.2–398.0 mm across sites from 2012 to 2016) (*Wang et al., 2020*). We established a 5-year (2012–2016) region-scale survey over this area (latitudes 39.34–49.96°N, longitudes 107.56–120.12°E), covering different grassland types (*Figure 1a*; *Wang et al., 2020*). There were 21 sites with 4–5 consecutive years of data. The sample plots of each site were randomly selected, excluding anthropogenic disturbances (e.g., overgrazing and heavy mowing). At each site, we marked three 1 × 1 m quadrats along the diagonal of a 10 × 10 m plot, harvested all living plant tissues, sorted them to species, and then oven-dried and weighed the harvested material to obtain aboveground biomass and species richness (for details, see *Wang et al., 2020*).

### Construction of regional communities

We constructed regional communities consisting of two local communities with a simulated landscape method (*Hautier et al., 2018*; *van der Plas et al., 2016*). Specifically, the 21 local communities (sites) were randomly separated into 10 regional communities without replacement (two local communities for each regional community with one remainder) to ensure that they were independent between each other (see *Figure 1b* for a simplified seven-site case). We repeated this random resampling process 1000 times, resulting in 1000 resampled sets, each containing 10 regional communities that were independent of each other.

### Temporal CV, synchrony, and species diversity across ecological hierarchies

A regional community includes $M$ local communities ($M$ = 2 for the current case) and $S$ species (see *Box 1* and *Appendix 1—table 1*). Its temporal dynamics can be described with a matrix with elements $u^{P,L}(i, k)$ for the mean abundance of species $k$ in locality $i$, and a matrix with elements $v^{P,L}(ij, kl) = cov(N^{P,L}(i, k, t), N^{P,L}(j, l, t))$ for the covariance between abundances of species $k$ in locality $i$ and species $l$ in locality $j$ over time $t$. Here, $N$ denotes population abundance, and the superscripts indicate 'population' ($P$) and 'local' ($L$). To estimate CV and synchrony with only dominant species (designated with subscript $d$, see Appendix 1 for detailed mathematical derivation), we introduce a matrix $d^P$ with elements $d^P(I, k)$ set to 1 if the species $k$ is a dominant species, otherwise, 0. We defined dominant species as those whose biomass contributed to >5% of the total biomass of the regional community (*Wang et al., 2020*) over the years of the survey (see *Appendix 1—figure 1*).

The local population CV can be upscaled to the regional community CV via two alterative pathways I or II (**Box 1**; **Wang et al., 2019**). First, we estimated the local population CV of all species ($CV^{P,L}$, **Equation 1a**) and only dominant species ($CV_d^{P,L}$, **Equation 1b**) as follows:

$$CV^{P,L} = \sum_{i,k} \frac{u^{P,L}(i,k)}{u^{C,R}} \frac{\sqrt{v^{P,L}(ii,kk)}}{u^{P,L}(i,k)} \tag{1a}$$

$$CV_d^{P,L} = \sum_{i,k} d^P(i,k) \frac{u^{P,L}(i,k)}{u^{C,R}} \frac{\sqrt{v^{P,L}(ii,kk)}}{u^{P,L}(i,k)} \tag{1b}$$

Here, $u^{C,R}$ is the average total biomass of the regional community over time and the superscripts indicate 'community' (*C*) and 'regional' (*R*). Second, we estimated the local species synchrony of all species ($\varphi^{P\to C,L}$, **Equation 2a**) and only dominant species ($\varphi_d^{P\to C,L}$, **Equation 2b**):

$$\varphi^{P\to C,L} = \sum_i \frac{\sum_k \sqrt{v^{P,L}(ii,kk)}}{\sum_{i,k} \sqrt{v^{P,L}(ii,kk)}} \frac{\sqrt{\sum_{kl} v^{P,L}(ii,kl)}}{\sum_k \sqrt{v^{P,L}(ii,kk)}} \tag{2a}$$

$$\varphi_d^{P\to C,L} = \sum_i \frac{\sum_k \sqrt{v^{P,L}(ii,kk)}}{\sum_{i,k} \sqrt{v^{P,L}(ii,kk)}} \frac{\sqrt{\sum_{kl} d^P(i,k) d^P(i,l) v^{P,L}(ii,kl)}}{\sum_k \sqrt{v^{P,L}(ii,kk)}} \tag{2b}$$

Here, the superscript ($P\to C$, *L*) indicates upscaling along pathway I, step A, from local population CV and local species synchrony to local community CV. Third, the local community CV of all species ($CV^{C,L}$, **Equation 3a**) and only dominant species ($CV_d^{C,L}$, **Equation 3b**) was estimated:

$$CV^{C,L} = \varphi^{P\to C,L} \times CV^{P,L} \tag{3a}$$

$$CV_d^{C,L} = \varphi_d^{P\to C,L} \times CV_d^{P,L} \tag{3b}$$

The regional community synchrony of all species ($\varphi^{C,L\to R}$, **Equation 4a**) and only dominant species ($\varphi_d^{C,L\to R}$, **Equation 4b**) can be estimated as follows:

$$\varphi^{C,L\to R} = \frac{\sqrt{\sum_{ij,kl} v^{P,L}(ij,kl)}}{\sum_i \sqrt{v^{C,L}(ii)}} \tag{4a}$$

$$\varphi_d^{C,L\to R} = \frac{\sqrt{\sum_{ij,kl} d^P(i,k) d^P(j,l) v^{P,L}(ij,kl)}}{\sum_i \sqrt{v^{C,L}(ii)}} \tag{4b}$$

The superscript (*C*, $L\to R$) indicates upscaling from local community CV and regional community synchrony to regional community CV (pathway I, step B). Here, $v^{C,L}(ii) = \sum_{kl} v^{P,L}(ii,kl)$ is the variance of community biomass at locality *i* over time. Finally, the regional community CV of all species ($CV^{C,R}$, **Equation 5a**) and only dominant species ($CV_{d\_C}^{C,R}$, **Equation 5b**) along the upscaling pathway was estimated:

$$CV^{C,R} = \varphi^{C,L\to R} \times CV^{C,L} = \varphi^{C,L\to R} \times \varphi^{P\to C,L} \times CV^{P,L} \tag{5a}$$

$$CV_{d\_C}^{C,R} = \varphi_d^{C,L\to R} \times CV_d^{C,L} = \varphi_d^{C,L\to R} \times \varphi_d^{P\to C,L} \times CV_d^{P,L} \tag{5b}$$

Note that the regional community CV of only dominant species is presented here only for completeness, but in our empirical analysis we were more interested in the relationship between the regional community CV of all species and dominant species dynamics. That is, we wanted to test the explanatory power of dominant-relative to all-species components in predicting the all-species regional community CV (i.e., $CV^{C,R}$; **Box 1**).

Along pathway II, local population CV and regional population synchrony scale up to regional population CV at first (pathway II, step A). The regional population synchrony of all species ($\varphi^{P,L\to R}$, **Equation 6a**) and only dominant species ($\varphi_d^{P,L\to R}$, **Equation 6b**) is estimated as follows:

$$\varphi^{P,L\to R} = \sum_k \frac{\sum_i \sqrt{v^{P,L}(ii,kk)}}{\sum_{i,k} \sqrt{v^{P,L}(ii,kk)}} \frac{\sqrt{\sum_{ij} v^{P,L}(ij,kk)}}{\sum_i \sqrt{v^{P,L}(ii,kk)}} \tag{6a}$$

$$\varphi_d^{P,L\to R} = \sum_k \frac{\sum_i \sqrt{v^{P,L}(ii,kk)}}{\sum_{i,k}\sqrt{v^{P,L}(ii,kk)}}\frac{\sqrt{\sum_{ij}d^P(i,k)d^P(j,k)v^{P,L}(ij,kk)}}{\sum_i\sqrt{v^{P,L}(ii,kk)}} \tag{6b}$$

The regional population CV of all species ($CV^{P,R}$, **Equation 7a**) and only dominant species ($CV_d^{P,R}$, **Equation 7b**) is

$$CV^{P,R} = \varphi^{P,L\to R} \times CV^{P,L} \tag{7a}$$

$$CV_d^{P,R} = \varphi_d^{P,L\to R} \times CV_d^{P,L} \tag{7b}$$

The regional population CV and regional species synchrony scale up to the regional community CV (pathway II, step B). The regional species synchrony of all species ($\varphi^{P\to C,R}$, **Equation 8a**) and only dominant species ($\varphi_d^{P\to C,R}$, **Equation 8b**) is estimated as follows:

$$\left(\varphi^{P\to C,R}\right)^2 = \frac{\sum_{ij,kl}v^{P,L}(ij,kl)}{\left(\sum_k\sqrt{v^{P,R}(kk)}\right)^2} \tag{8a}$$

$$\left(\varphi_d^{P\to C,R}\right)^2 = \frac{\sum_{ij,kl}d^P(i,k)d^P(j,l)v^{P,L}(ij,kl)}{\left(\sum_k\sqrt{v^{P,R}(kk)}\right)^2} \tag{8b}$$

Here, $v^{P,R}(kk)$ is the variance of population biomass of species $k$ at the regional scale over time. The regional community CV of all species ($CV^{C,R}$, **Equation 9a**) and only dominant species ($CV_{d\_P}^{C,R}$, **Equation 9b**) according to the upscaling pathway II can be estimated as follows:

$$CV^{C,R} = \varphi^{P\to C,R} \times CV^{P,R} = \varphi^{P\to C,R} \times \varphi^{P,L\to R} \times CV^{P,L} \tag{9a}$$

$$CV_{d\_P}^{C,R} = \varphi_d^{P\to C,R} \times CV_d^{P,R} = \varphi_d^{P\to C,R} \times \varphi_d^{P,L\to R} \times CV_d^{P,L} \tag{9b}$$

Here again the regional community CV of only dominant species is listed for completeness, but the regional community CV of all species was related to dominant species dynamics to test the explanatory power of dominant-relative to all-species components in predicting the all-species regional community CV along upscaling pathway II (see **Box 1**).

The regional community CV estimated with the two alterative upscaling pathways is the same when using all-species measures, but can be slightly different when using dominant-species measures (see Appendix 1–1.6 for details), which is why we used two abbreviations, that is, $CV_{d\_C}^{C,R}$ and $CV_{d\_P}^{C,R}$.

We estimated two alternative species diversity indices across ecological hierarchies, species richness ($S$) and effective species richness ($D$). The alpha ($S^\alpha$) and gamma species richness ($S^\gamma$) were defined as the total number of species at local and regional scales, respectively, and the beta species richness ($S^\beta=S^\gamma/S^\alpha$) was used to measure dissimilarity among localities. Specifically, the alpha ($S^\alpha$) and gamma ($S^\gamma$) species richness were estimated as multiple-year mean ($S^\alpha$) and multiple-year pooled species number ($S^\gamma$) of the two local communities. To account for uneven species abundances in the study region, we also used effective species richness, the antilog of Shannon–Wiener diversity ($D = e^H$), reflecting how many species with an even abundance distribution would produce the same Shannon–Wiener diversity as observed for the actual uneven community (**Wang et al., 2020**). The alpha ($D^\alpha$) and gamma ($D^\gamma$) effective species richness thus represented the Shannon–Wiener diversity at local and regional scales, respectively, with beta effective species richness ($D^\beta = D^\gamma/D^\alpha$) measuring its cross-locality dissimilarity. These species diversity indices were estimated with either all species or only dominant species.

## Climatic data

Based on monthly climatic data collected from 119 climate stations and 2 km resolution digital elevation over this region, we calculated site-specific mean temperature and precipitation using the simple kriging method and spherical model of geostatistical analysis in ArcGIS software (Environmental Systems Research Institute Inc, Redlands, CA). We calculated mean annual temperature (*MAT*) and annual precipitation (*MAP*) (only monthly data from May to October were used as plants are active only in this period), as well as their CVs across spatial scales from 2012 to 2016. CVs of temperature and precipitation at the local ($CV_T^L$ and $CV_P^L$) and regional scales ($CV_T^R$ and $CV_P^R$), as well as their

among-site synchronies ($\varphi_T{}^{L \to R}$ and $\varphi_P{}^{L \to R}$), were estimated with the methods used for local and regional community CVs and regional community synchrony.

## Statistical analysis

We analyzed the influence of distance between spatially separated local communities (i.e., sites) within regional communities on regional synchronies of temperature and precipitation, regional community CV, as well as all-species and dominant-species measures of regional population CV and synchrony and regional community synchrony with linear regressions. However, spatial distance only influenced regional synchronies of temperature and precipitation (*Figure 1—figure supplement 1*), and thus distance was not considered further in subsequent analyses.

We used correlation analyses, linear regression analyses, and path analyses to investigate the regional community CV in relation to its hierarchical components, species diversity indices, and climatic factors. For the path analyses, we established models considering different upscaling pathways and different species diversity indices to explain variation in the all-species regional community CV either using all species or only dominant species. We started with initial models as close as possible to paths proposed in theoretical studies (*Wang et al., 2019*; *Wang and Loreau, 2016*; *Wang and Loreau, 2014*). GLMs were used to analyze the proposed paths. Then, we modified the initial path-analysis models (*Appendix 1—figure 4* and *Appendix 1—figure 5*) to eliminate nonsignificant paths until only significant or marginally significant paths remained or a minimal value of Akaike's information criterion for small sample size (*AICc*) was reached (*Brewer et al., 2016*). Path coefficients of the final models were quantified using the *piecewiseSEM* package (*Lefcheck, 2016*) of R 3.6.3 (*R Development Core Team, 2013*).

We used a randomized examination method to investigate the statistical significance of the above analyses (*Edgington and Onghena, 2007*; *Efron and Tibshirani, 1994*). Specifically, considering the 10 independent regional communities per sampled set, all above statistical analyses were conducted within each set, resulting in 1000 statistics. Taking the correlation analysis as an example, we calculated the mean correlation coefficient ($\bar{\rho}$) of the 1000 sets and considered it to be statistically significant or marginally significant if the proportion of $\rho < 0$ ($P_{-\rho}$) (or $\rho > 0$, $P_{+\rho}$) was lower than 0.05 or 0.10 when $\bar{\rho} > 0$ (or $\bar{\rho} < 0$), respectively.

## Acknowledgements

This study was supported by the National Nature Science Foundation of China (31960259, 31971434, 32160274, 31370454, and 31600385), the Strategic Priority Research Program of Chinese Academy of Sciences (XDA26010101), the National Key Research and Development Program of China (2016YFC0500602), the Ministry of Science and Technology of China (2015BAC02B04), and the Natural Science Foundation of Inner Mongolia (2019MS03089, 2019MS03088, and 2015ZD05). SW was supported by the National Nature Science Foundation of China (31988102). BS was supported by the University Research Priority Program Global Change and Biodiversity of the University of Zurich. All authors declare no conflict of interest.

## Additional information

### Funding

| Funder | Grant reference number | Author |
| --- | --- | --- |
| National Natural Science Foundation of China | 31960259 | Wenhong Ma |
| National Natural Science Foundation of China | 31971434 | Yonghui Wang |
| National Natural Science Foundation of China | 32160274 | Yonghui Wang |
| National Natural Science Foundation of China | 31370454 | Wenhong Ma |

| Funder | Grant reference number | Author |
|---|---|---|
| National Natural Science Foundation of China | 31600385 | Yonghui Wang |
| Chinese Academy of Sciences | Strategic Priority Research Program XDA26010101 | Yonghui Wang |
| National Key Research and Development Program of China | 2016YFC0500602 | Yonghui Wang |
| Ministry of Science and Technology of the People's Republic of China | 2015BAC02B04 | Yonghui Wang |
| Natural Science Foundation of Inner Mongolia | 2019MS03089 | Wenhong Ma |
| Natural Science Foundation of Inner Mongolia | 2019MS03088 | Yonghui Wang |
| Natural Science Foundation of Inner Mongolia | 2015ZD05 | Yonghui Wang |
| National Natural Science Foundation of China | 31988102 | Shaopeng Wang |
| University of Zurich | University Research Priority Program Global Change and Biodiversity | Bernhard Schmid |

The funders had no role in study design, data collection and interpretation, or the decision to submit the work for publication.

#### Author contributions

Yonghui Wang, Conceptualization, Data curation, Funding acquisition, Methodology, Writing – original draft, Writing – review and editing; Shaopeng Wang, Bernhard Schmid, Conceptualization, Funding acquisition, Methodology, Writing – original draft, Writing – review and editing; Liqing Zhao, Cunzhu Liang, Bailing Miao, Qing Zhang, Xiaxia Niu, Data curation; Wenhong Ma, Conceptualization, Data curation, Funding acquisition, Writing – original draft, Writing – review and editing

#### Author ORCIDs

Yonghui Wang ⓘ http://orcid.org/0000-0002-3351-4134
Bernhard Schmid ⓘ http://orcid.org/0000-0002-8430-3214

#### Decision letter and Author response

Decision letter https://doi.org/10.7554/eLife.74881.sa1
Author response https://doi.org/10.7554/eLife.74881.sa2

---

## Additional files

#### Supplementary files

• Transparent reporting form

#### Data availability

The data that support the findings of this study are openly available in Figshare at https://doi.org/10.6084/m9.figshare.20281902.

The following dataset was generated:

| Author(s) | Year | Dataset title | Dataset URL | Database and Identifier |
|---|---|---|---|---|
| Wang Y | 2022 | Stability and asynchrony of local communities but less so diversity increase regional stability of Inner Mongolian grassland | https://doi.org/10.6084/m9.figshare.20281902 | figshare, 10.6084/m9.figshare.20281902 |

The following previously published dataset was used:

| Author(s) | Year | Dataset title | Dataset URL | Database and Identifier |
|---|---|---|---|---|
| Wang Y, Niu X, Zhao L, Liang C, Miao B, Zhang Q, Zhang J, Schmid B, Ma W | 2020 | Data from: Biotic stability mechanisms in Inner Mongolian grassland | https://doi.org/10.5061/dryad.ht76hdrc5 | Dryad Digital Repository, 10.5061/dryad.ht76hdrc5 |

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

# Appendix 1

## Mathematical derivation for partitioning coefficients of variation and synchronies across ecological hierarchies into dominant and subdominant species groups

In the following, we introduce methods partitioning (temporal) coefficients of variation (CVs, inverse of (temporal) stability) and synchronies (inverse of asynchrony) across ecological hierarchies into dominant (relative species abundance >5%, see *Appendix 1—figure 1* for details) and subdominant species groups. We do not repeat theoretical derivations relating CVs and synchronies across different hierarchies here because these can be found elsewhere (*Loreau and de Mazancourt, 2008*; *Thibaut and Connolly, 2013*; *Wang et al., 2019*; *Wang and Loreau, 2016*; *Wang and Loreau, 2014*). Symbols and descriptions used in the following partitions can be found in *Box 1* and *Appendix 1—table 1*.

**Appendix 1—table 1.** Notation summary for climatic factors, species diversity indices, coefficients of variation (CVs, inverse of stability), and synchronies (inverse of asynchrony) across spatial scales and hierarchical levels of ecological organization.

| Symbol | Description |
|---|---|
| *Climatic factors* | |
| $MAT$ | Cross-site averaged mean annual temperature |
| $MAP$ | Cross-site averaged mean annual precipitation |
| $CV_T^L$ | Local CV of temperature |
| $CV_P^L$ | Local CV of precipitation |
| $\varphi_T^{L \to R}$ | Regional temperature synchrony |
| $\varphi_P^{L \to R}$ | Regional precipitation synchrony |
| $CV_T^R$ | Regional CV of temperature |
| $CV_P^R$ | Regional CV of precipitation |
| *Biodiversity indices* | |
| $S^\alpha$ or $S_d^\alpha$ | Alpha species richness estimated with all species or only dominant species |
| $S^\beta$ or $S_d^\beta$ | Beta species richness estimated with all species or only dominant species |
| $S^\gamma$ or $S_d^\gamma$ | Gamma species richness estimated with all species or only dominant species |
| $D^\alpha$ or $D_d^\alpha$ | Alpha effective species richness estimated with all species or only dominant species |
| $D^\beta$ or $D_d^\beta$ | Beta effective species richness estimated with all species or only dominant species |
| $D^\gamma$ or $D_d^\gamma$ | Gamma effective species richness estimated with all species or only dominant species |
| *Stability and synchrony* | |
| $CV^{P,L}$ or $CV_d^{P,L}$ | Local population CV estimated with all species or only dominant species |
| $\varphi^{P \to C,L}$ or $\varphi_d^{P \to C,L}$ | Local species synchrony estimated with all species or only dominant species |
| $CV^{C,L}$ or $CV_d^{C,L}$ | Local community CV estimated with all species or only dominant species |
| $\varphi^{C,L \to R}$ or $\varphi_d^{C,L \to R}$ | Regional community synchrony estimated with all species or only dominant species |

*Appendix 1—table 1 Continued on next page*

*Appendix 1—table 1 Continued*

| Symbol | Description |
|---|---|
| $\varphi^{P,L \rightarrow R}$ or $\varphi_d{}^{P,L \rightarrow R}$ | Regional population synchrony estimated with all species or only dominant species |
| $CV^{P,R}$ or $CV_d{}^{P,R}$ | Regional population CV estimated with all species or only dominant species |
| $\varphi^{P \rightarrow C,R}$ or $\varphi_d{}^{P \rightarrow C,R}$ | Regional species synchrony estimated with all species or only dominant species |
| $CV^{C,R}$ or $CV_{d\_C}{}^{C,R}$ and $CV_{d\_P}{}^{C,R}$ | Regional community CV estimated with all species or only dominant species along pathways of aggregating local communities (pathway I: $CV_{d\_C}{}^{C,R} = \varphi_d{}^{C,L \rightarrow R} \times CV_d{}^{C,L}$) or organizing regional populations (pathway II: $CV_{d\_P}{}^{C,R} = \varphi_d{}^{P \rightarrow C,R} \times CV_d{}^{P,R}$) |

We consider a regional community reached a stationary state, which includes $M$ localities (e.g., sites or local communities) and $S$ species. This regional community can be described with a matrix of (temporal) mean species abundance with elements $u^{P,L}(i, k)$, that is, the mean abundance of species $k$ in locality $i$, and a (temporal) variance–covariance matrix of species abundances with elements $v^{P,L}(ij, kl) = cov(N^{P,L}(i, k, t), N^{P,L}(j, l, t))$, that is, the covariance between abundances of species $k$ in locality $i$ and species $l$ in locality $j$ over time $t$. In addition, we introduce two matrixes, $d^P$ and $s^P$, to represent the dominant and subdominant species of the regional community, respectively. For the $d^P$, it has $M$ rows and $S$ columns, representing numbers of locality and species of the regional community, and has elements $d^P(i, k)$, that is, the $k$th species of the $i$th locality, which is set to 1 if the $k$th species is a dominant species at the regional scale, otherwise, 0. Similar procedure is used to conduct the $s^P$, in which subdominant species are set to 1, otherwise, 0.

## 1.1 Partitioning local population CV into dominant and subdominant species groups

The local population CV ($CV^{P,L}$) is defined as the weighted-average local population CV (*Thibaut and Connolly, 2013*; *Wang et al., 2019*; *Wang and Loreau, 2016*; *Wang and Loreau, 2014*):

$$CV^{P,L} = \frac{\sum_{i,k} \sqrt{v^{P,L}(ii,kk)}}{u^{C,R}} = \sum_{i,k} \frac{u^{P,L}(i,k)}{u^{C,R}} \frac{\sqrt{v^{P,L}(ii,kk)}}{u^{P,L}(i,k)} \tag{S1}$$

We rewrite this equation with introduced two matrixes ($d^P(i, k)$ and $s^P(i, k)$) to separate the local population CV ($CV^{P,L}$) into its dominant ($CV_d{}^{P,L}$) and subdominant ($CV_s{}^{P,L}$) species group components:

$$\begin{aligned} CV^{P,L} &= \sum_{i,k} d^P(i,k) \frac{u^{P,L}(i,k)}{u^{C,R}} \frac{\sqrt{v^{P,L}(ii,kk)}}{u^{P,L}(i,k)} + \sum_{i,k} s^P(i,k) \frac{u^{P,L}(i,k)}{u^{C,R}} \frac{\sqrt{v^{P,L}(ii,kk)}}{u^{P,L}(i,k)} \\ &= CV_d^{P,L} + CV_s^{P,L} \end{aligned} \tag{S2}$$

## 1.2 Partitioning local species synchrony into dominant and subdominant species groups

The local species synchrony ($\varphi^{P \rightarrow C,L}$) is defined as the weighted-average synchronous dynamics among populations of different species within local communities (*Wang et al., 2019*; *Wang and Loreau, 2016*; *Wang and Loreau, 2014*):

$$\varphi^{P \rightarrow C,L} = \frac{\sum_i \sqrt{v^{C,L}(ii)}}{\sum_{i,k} \sqrt{v^{P,L}(ii,kk)}} = \sum_i \frac{\sum_k \sqrt{v^{P,L}(ii,kk)}}{\sum_{i,k} \sqrt{v^{P,L}(ii,kk)}} \frac{\sqrt{\sum_{kl} v^{P,L}(ii,kl)}}{\sum_k \sqrt{v^{P,L}(ii,kk)}} = \sum_i \omega^{P \rightarrow C,L}(i)\, \varphi^{P \rightarrow C,L}(i) \tag{S3}$$

where $\omega^{P \rightarrow C,L}(i)$ and $\varphi^{P \rightarrow C,L}(i)$ are the contribution of local population variance of the $i$th community to the sum of variance of all species local populations within the regional community and synchronous dynamics among local populations of different species within the $i$th local community (i.e., species synchrony of the $i$th local community, *Loreau and de Mazancourt, 2008*), respectively. We can rewrite $\varphi^{P \rightarrow C,L}(i)$ with $d^P(i, k)$ and $s^P(i, k)$:

$$\left(\varphi^{P\to C,L}(i)\right)^2 \;=\; \frac{\sum_{kl} v^{P,L}(ii,kl)}{\left(\sum_k \sqrt{v^{P,L}(ii,kk)}\right)^2} \;=\; \frac{\sum_{kl} d^P(i,k)\,d^P(i,l)\,v^{P,L}(ii,kl)}{\left(\sum_k \sqrt{v^{P,L}(ii,kk)}\right)^2} \;+\; \frac{2\sum_{kl} d^P(i,k)\,s^P(i,l)\,v^{P,L}(ii,kl)}{\left(\sum_k \sqrt{v^{P,L}(ii,kk)}\right)^2} \;+\;$$

$$\frac{\sum_{kl} s^P(i,k)\,s^P(i,l)\,v^{P,L}(ii,kl)}{\left(\sum_k \sqrt{v^{P,L}(ii,kk)}\right)^2}$$

(S4)

We define the first term of the right-hand side of the *Equation S4* as the dominant-species local species synchrony of the *i*th local community (*Wang et al., 2020*):

$$\varphi_d^{P\to C,L}(i) \;=\; \frac{\sqrt{\sum_{kl} d^P(i,k)\,d^P(i,l)\,v^{P,L}(ii,kl)}}{\sum_k \sqrt{v^{P,L}(ii,kk)}}$$

(S5)

Then, using above description, we define the dominant-species local species synchrony of the regional community ($\varphi_d^{P\to C,L}$), that is, an aggregation of multiple local communities:

$$\varphi_d^{P\to C,L} \;=\; \sum_i \omega^{P\to C,L}(i)\,\varphi_d^{P\to C,L}(i)$$

(S6)

Referring to the definition of local community CV, $CV^{C,L} = \varphi^{P\to C,L} \times CV^{P,L}$ (*Wang et al., 2019*; *Wang and Loreau, 2016*; *Wang and Loreau, 2014*), we define the dominant-species local community CV ($CV_d^{C,L}$):

$$CV_d^{C,L} \;=\; \varphi_d^{P\to C,L} \times CV_d^{P,L}$$

(S7)

## 1.3 Partitioning regional community synchrony into dominant and subdominant species groups

The regional community synchrony ($\varphi^{C,L\to R}$) is defined as the weighted-average synchronous dynamics among spatially separated local communities (*Wang et al., 2019*; *Wang and Loreau, 2016*; *Wang and Loreau, 2014*):

$$\left(\varphi^{C,L\to R}\right)^2 \;=\; \frac{\sum_{ij} v^{C,L}(ij)}{\left(\sum_i \sqrt{v^{C,L}(ii)}\right)^2} \;=\; \frac{\sum_{ij,kl} v^{P,L}(ij,kl)}{\left(\sum_i \sqrt{v^{C,L}(ii)}\right)^2}$$

(S8)

Using $d^P(i,k)$ and $s^P(i,k)$ mentioned above, we partition regional community synchrony into dominant ($\varphi_d^{C,L\to A}$), subdominant species groups ($\varphi_s^{C,L\to A}$), and synchronous dynamic between them ($\varphi_{ds}^{C,L\to A}$):

$$\left(\varphi^{C,L\to R}\right)^2 \;=\; \frac{\sum_{ij,kl} d^P(i,k)\,d^P(j,l)\,v^{P,L}(ij,kl)}{\left(\sum_i \sqrt{v^{C,L}(ii)}\right)^2} + \frac{2\sum_{ij,kl} d^P(i,k)\,s^P(j,l)\,v^{P,L}(ij,kl)}{\left(\sum_i \sqrt{v^{C,L}(ii)}\right)^2} + \frac{\sum_{ij,kl} s^P(i,k)\,s^P(j,l)\,v^{P,L}(ij,kl)}{\left(\sum_i \sqrt{v^{C,L}(ii)}\right)^2}$$

$$=\; \left(\varphi_d^{C,L\to R}\right)^2 + \left(\varphi_{ds}^{C,L\to R}\right)^2 + \left(\varphi_s^{C,L\to R}\right)^2$$

(S9)

Referring to the definition of regional community CV with the upscaling pathway of aggregating local communities (pathway I), $CV^{C,R} = \varphi^{C,L\to R} \times CV^{C,L}$ (*Wang et al., 2019*; *Wang and Loreau, 2016*; *Wang and Loreau, 2014*), we define the dominant-species regional community CV with this upscaling pathway ($CV_{d\_C}^{C,R}$):

$$CV_{d\_C}^{C,R} \;=\; \varphi_d^{C,L\to R} \times CV_d^{C,L} \;=\; \varphi_d^{C,L\to R} \times \varphi_d^{P\to C,L} \times CV_d^{P,L}$$

(S10)

## 1.4 Partitioning regional population synchrony into dominant and subdominant species groups

The regional population synchrony ($\varphi^{P,L\to R}$) is defined as the weighted-average synchronous dynamics among spatially separated local populations of same species (*Wang et al., 2019*):

$$\varphi^{P,L\to R} \;=\; \frac{\sum_k \sqrt{v^{P,R}(kk)}}{\sum_{i,k} \sqrt{v^{P,L}(ii,kk)}} \;=\; \sum_k \frac{\sum_i \sqrt{v^{P,L}(ii,kk)}}{\sum_{i,k} \sqrt{v^{P,L}(ii,kk)}} \frac{\sqrt{\sum_{ij} v^{P,L}(ij,kk)}}{\sum_i \sqrt{v^{P,L}(ii,kk)}} \;=\; \sum_k \omega^{P,L\to R}(k)\,\varphi^{P,L\to R}(k)$$

(S11)

where $\omega^{P,L \to R}(k)$ and $\varphi^{P,L \to R}(k)$ are the contributions of population variance of the $k$th species to that of all species within the regional community and synchrony within the $k$th species among sites, respectively. We can rewrite $\varphi^{P,L \to R}(k)$ with $d^P(i, k)$ and $s^P(i, k)$:

$$\left(\varphi^{P,L \to R}(k)\right)^2 = \frac{\sum_{ij} v^{P,L}(ij,kk)}{\left(\sum_i \sqrt{v^{P,L}(ii,kk)}\right)^2} = \frac{\sum_{ij} d^P(i,k) d^P(j,k) v^{P,L}(ij,kk)}{\left(\sum_i \sqrt{v^{P,L}(ii,kk)}\right)^2} + \frac{2 \sum_{ij} d^P(i,k) s^P(j,k) v^{P,L}(ij,kk)}{\left(\sum_i \sqrt{v^{P,L}(ii,kk)}\right)^2} +$$

$$\frac{\sum_{ij} s^P(i,k) s^P(j,k) v^{P,L}(ij,kk)}{\left(\sum_i \sqrt{v^{P,L}(ii,kk)}\right)^2}$$

(S12)

We define the first term of the right-hand side of above equation as the regional population synchrony of the $k$th (dominant) species:

$$\varphi_d^{P,L \to R}(k) = \frac{\sqrt{\sum_{ij} d^P(i,k) d^P(j,k) v^{P,L}(ij,kk)}}{\sum_i \sqrt{v^{P,L}(ii,kk)}}$$

(S13)

Then, using the above description, we defined the dominant-species estimate of regional population synchrony ($\varphi_d^{P,L \to R}$):

$$\varphi_d^{P,L \to R} = \sum_k \omega^{P,L \to R}(k) \, \varphi_d^{P,L \to R}(k)$$

(S14)

Referring to the definition of regional population CV, $CV^{P,R} = \varphi^{P,L \to R} \times CV^{P,L}$ (**Wang et al., 2019**), we define the dominant-species regional population CV ($CV_d^{P,R}$):

$$CV_d^{P,R} = \varphi_d^{P,L \to R} \times CV_d^{P,L}$$

(S15)

## 1.5 Partitioning regional species synchrony into dominant and subdominant species groups

The regional species synchrony ($\varphi^{P \to C,R}$) is defined as the weighted-average synchronous dynamics among regional populations of different species (**Wang et al., 2019**):

$$\left(\varphi^{P \to C,R}\right)^2 = \frac{\sum_{kl} v^{P,R}(kl)}{\left(\sum_k \sqrt{v^{P,R}(kk)}\right)^2} = \frac{\sum_{ij,kl} v^{P,L}(ij,kl)}{\left(\sum_k \sqrt{v^{P,R}(kk)}\right)^2}$$

(S16)

Here, $v^{P,R}(kl)$ is the covariance between $k$ and $l$ regional populations. We partition the regional species synchrony into dominant ($\varphi_d^{S \to C,R}$), subdominant species groups ($\varphi_s^{S \to C,R}$), and synchronous dynamic between them ($\varphi_{ds}^{S \to C,R}$) using introduced $d^P(i, k)$ and $s^P(i, k)$:

$$\left(\varphi^{P \to C,R}\right)^2 = \frac{\sum_{ij,kl} v^{P,L}(ij,kl)}{\left(\sum_k \sqrt{v^{P,R}(kk)}\right)^2} = \frac{\sum_{ij,kl} d^P(i,k) d^P(j,l) v^{P,L}(ij,kl)}{\left(\sum_k \sqrt{v^{P,R}(kk)}\right)^2} + \frac{2 \sum_{ij,kl} d^P(i,k) s^P(j,l) v^{P,L}(ij,kl)}{\left(\sum_k \sqrt{v^{P,R}(kk)}\right)^2} +$$

$$\frac{\sum_{ij,kl} s^P(i,k) s^P(j,l) v^{P,L}(ij,kl)}{\left(\sum_k \sqrt{v^{P,R}(kk)}\right)^2} = \left(\varphi_d^{P \to C,R}\right)^2 + \left(\varphi_{ds}^{P \to C,R}\right)^2 + \left(\varphi_s^{P \to C,R}\right)^2$$

(S17)

Referring to the definition of regional community CV with the upscaling pathway of organizing regional populations (pathway II), $CV^{C,R} = \varphi^{P \to C,R} \times CV^{P,R}$ (**Wang et al., 2019**), we define the dominant-species regional community CV with this upscaling pathway ($CV_{d\_P}^{C,R}$):

$$CV_{d\_P}^{C,R} = \varphi_d^{P \to C,R} \times CV_d^{P,R} = \varphi_d^{P \to C,R} \times \varphi_d^{P,L \to R} \times CV_d^{P,L}$$

(S18)

## 1.6 Comparing dominant-species regional community CVs estimated with two alternative upscaling pathways

Based on a recent theoretical study (**Wang et al., 2019**), the regional community CV can be upscaled by aggregating local communities ($CV_C^{C,R}$) or organizing regional populations ($CV_P^{C,R}$):

$$CV_C^{C,R} = \varphi^{C,L\to R} \times \varphi^{P\to C,L} \times CV^{P,L}$$

$$= \frac{\sqrt{\sum_{ij,kl} v^{P,L}(ij,kl)}}{\sum_i \sqrt{v^{C,L}(ii)}} \times \frac{\sum_i \sqrt{v^{C,L}(ii)}}{\sum_{i,k} \sqrt{v^{P,L}(ii,kk)}} \times \frac{\sum_{i,k} \sqrt{v^{P,L}(ii,kk)}}{u^{C,R}} = \frac{\sqrt{\sum_{ij,kl} v^{P,L}(ij,kl)}}{u^{C,R}} \tag{S19}$$

$$CV_P^{C,R} = \varphi^{P\to C,R} \times \varphi^{P,\,L\to R} \times CV^{P,L}$$

$$= \frac{\sqrt{\sum_{ij,kl} v^{P,L}(ij,kl)}}{\sum_k \sqrt{v^{P,R}(kk)}} \times \frac{\sum_k \sqrt{v^{P,R}(kk)}}{\sum_{i,k} \sqrt{v^{P,L}(ii,kk)}} \times \frac{\sum_{i,k} \sqrt{v^{P,L}(ii,kk)}}{u^{C,R}} = \frac{\sqrt{\sum_{ij,kl} v^{P,L}(ij,kl)}}{u^{C,R}} \tag{S20}$$

These descriptions (*Equations S19 and S20*) show that the regional community CVs estimated with two different upscaling pathways are equal to each other.

In the following part, we explain why the dominant-species regional community CV estimated with two different upscaling pathways are not equal to each other ($CV_{d\_C}^{C,R}$ for estimated via aggregating local communities, pathway I, and $CV_{d\_P}^{C,R}$ for estimated via organizing regional populations, pathway II). The dominant-species regional community CV estimated by aggregating local communities ($CV_{d\_C}^{C,R}$) is

$$CV_{d\_C}^{C,R} = \varphi_d^{C,L\to R} \times \varphi_d^{P\to C,L} \times CV_d^{P,L} = \frac{\sqrt{\sum_{ij,kl} d^P(i,k) d^P(j,l) v^{P,L}(ij,kl)}}{\sum_i \sqrt{v^{C,L}(ii)}} \times \frac{\sum_i \sqrt{\sum_{kl} d^P(i,k) d^P(i,l) v^{P,L}(ii,kl)}}{\sum_{i,k} \sqrt{v^{P,L}(ii,kk)}}$$

$$\times \frac{\sum_{i,k} d^P(i,k) \sqrt{v^{P,L}(ii,kk)}}{u^{C,R}} \tag{S21}$$

The dominant-species regional community CV estimated by organizing regional populations ($CV_{d\_P}^{C,R}$) has the following description:

$$CV_{d\_P}^{C,R} = \varphi_d^{P\to C,R} \times \varphi_d^{P,L\to R} \times CV_d^{P,L} = \frac{\sqrt{\sum_{ij,kl} d^P(i,k) d^P(j,l) v^{P,L}(ij,kl)}}{\sum_k \sqrt{v^{P,R}(kk)}} \times \frac{\sum_k \sqrt{\sum_{ij} d^P(i,k) d^P(j,k) v^{P,L}(ij,kk)}}{\sum_{i,k} \sqrt{v^{P,L}(ii,kk)}}$$

$$\times \frac{\sum_{i,k} d^P(i,k) \sqrt{v^{P,L}(ii,kk)}}{u^{C,R}} \tag{S22}$$

Because these two equations have either same terms or different terms ($\frac{\sum_i \sqrt{\sum_{kl} d^P(i,k) d^P(i,l) v^{P,L}(ii,kl)}}{\sum_i \sqrt{v^{C,L}(ii)}}$ in *Equation S21* and $\frac{\sum_k \sqrt{\sum_{ij} d^P(i,k) d^P(j,k) v^{P,L}(ij,kk)}}{\sum_k \sqrt{v^{P,R}(kk)}}$ in *Equation S22*), the dominant-species regional community CV estimated with two different upscaling pathways should be well correlated but not exactly the same. The denominators of these two different terms are the sum of local community variances or the sum of regional population variances, respectively. The numerators are the sum of variances (and covariances) of different dominant species within the same local communities and the sum of variances (and covariances) of the same dominant species across different local communities, respectively. These differences reflect that dominant-species regional community CVs estimated via aggregating local communities ($CV_{d\_C}^{C,R}$) and via organizing regional populations ($CV_{d\_P}^{C,R}$) focus on different dominant species within the same local communities and the same dominant species across different local communities, respectively. Owing to the potential difference, we separately reported them (*Figure 3—figure supplement 1a and b*). We also note that the different terms in *Equation S21* and *Equation S22* can be same when considering all species. This is because, in this case, they become to $\frac{\sum_i \sqrt{\sum_{kl} v^{P,L}(ii,kl)}}{\sum_i \sqrt{v^{C,L}(ii)}}$ and $\frac{\sum_k \sqrt{\sum_{ij} v^{P,L}(ij,kk)}}{\sum_k \sqrt{v^{P,A}(kk)}}$ , and both of them are equal to 1, resulting in the same regional community CV estimated with all species using different upscaling pathways.

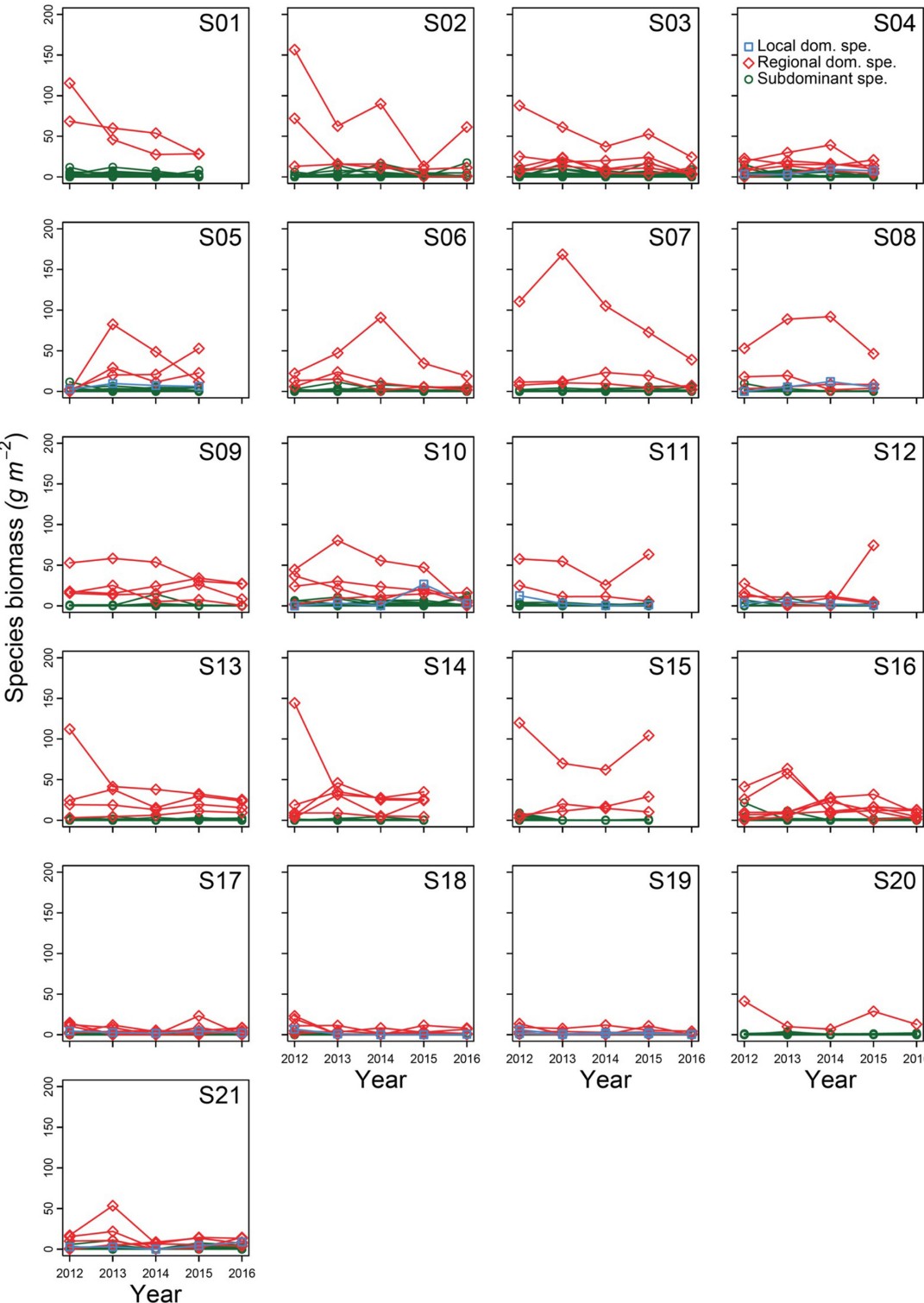

**Appendix 1—figure 1.** Time series of plant species biomass in each surveyed site. Blue squares and lines represent species that only characterized as dominant species in local communities. Red diamonds and lines represent species characterized as dominant species in local communities and can also be characterized as dominant species when aggregating into regional communities. Green circles and lines represent subdominant species. It showed that most dominant species of local communities can be defined as dominant species of regional communities, with only a few exceptions. In addition, these species have higher productivity than others roughly all the time and are constantly exist in surveyed sites. Dataset and code are available in Figshare at https://doi.org/10.6084/m9.figshare.20281902.

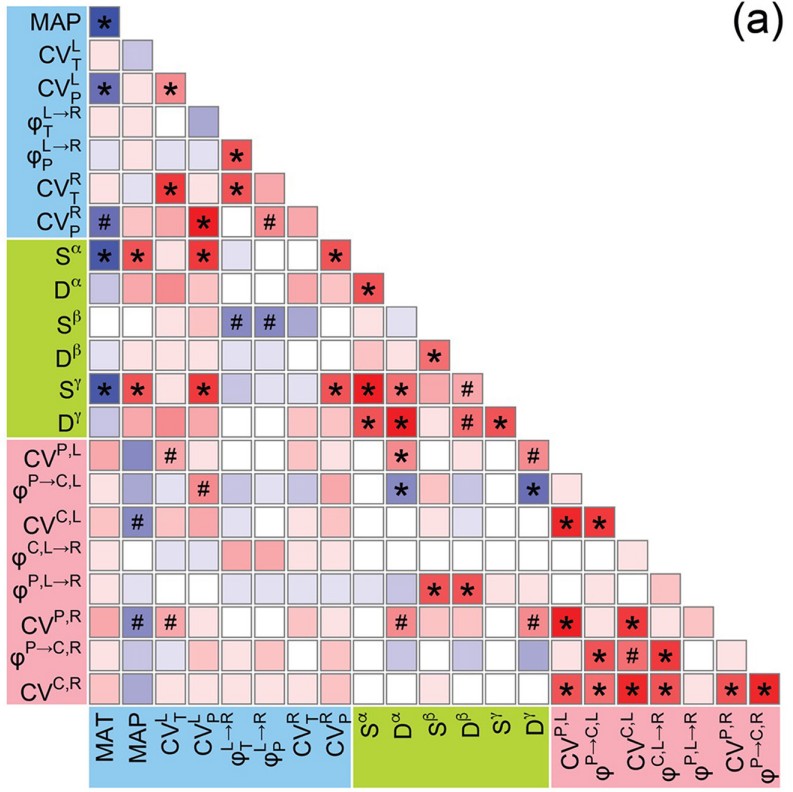

(a)

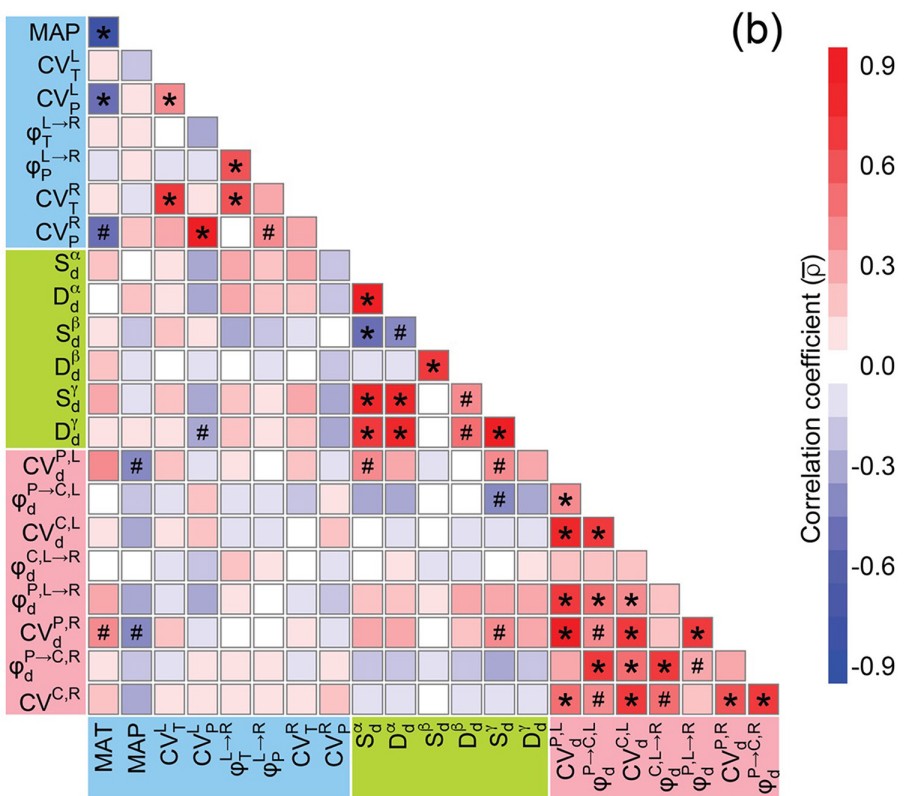

(b)

**Appendix 1—figure 2.** Correlation matrices for climatic factors, species diversity indices, coefficients of variation (CVs, inverse of stabilities), and synchronies (inverse of asynchronies) estimated with all species (**a**) and only

*Appendix 1—figure 2 continued on next page*

*Appendix 1—figure 2 continued*

dominant species (**b**) by considering a two-local-community scenario (see *Figure 1b* for a simplified case and *Appendix 1—figure 3* for a three-local-community scenario). Significant and marginally significant correlations are marked with *p<0.05 and #p<0.10, respectively (see 'Materials and methods' for details). Symbols and descriptions can be found in *Box 1* and *Appendix 1—table 1*. Dataset, code, and relevant results are available in Figshare at https://doi.org/10.6084/m9.figshare.20281902.

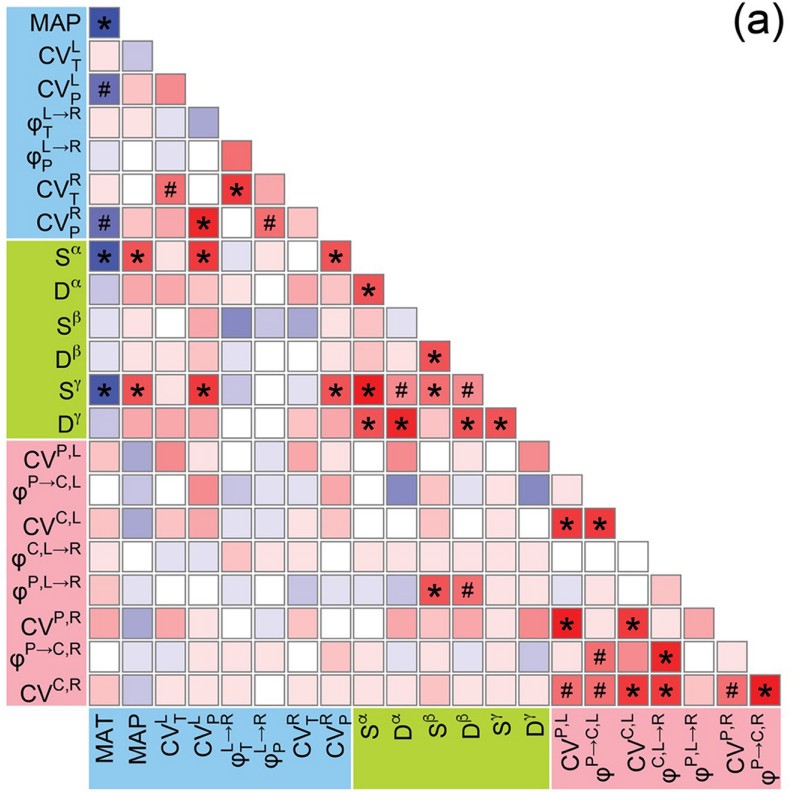

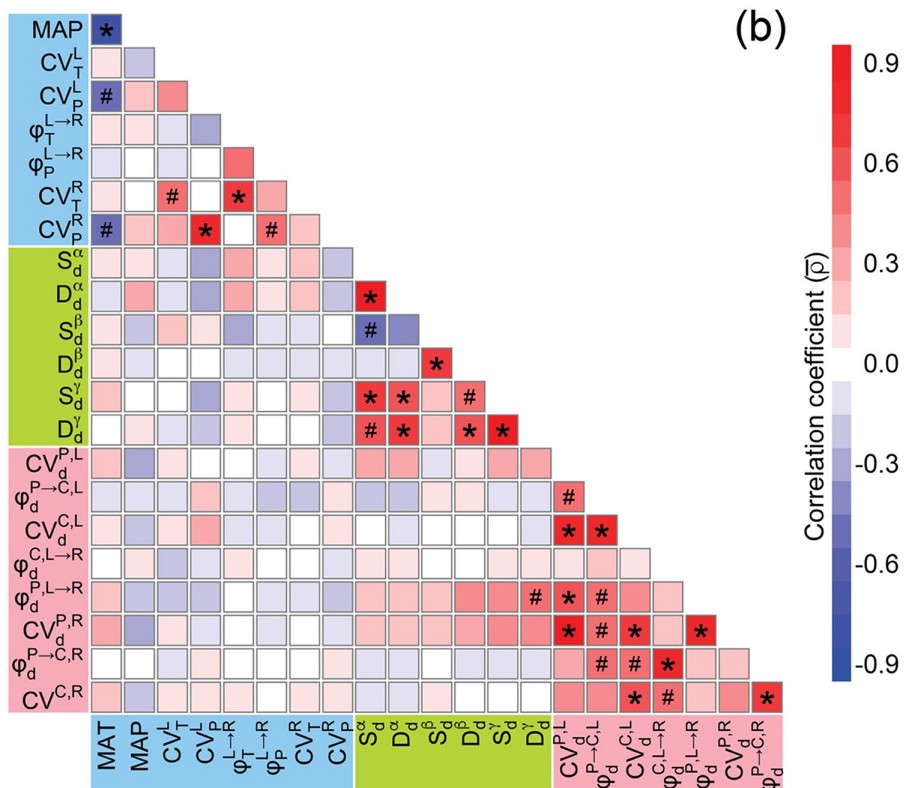

**Appendix 1—figure 3.** Correlation matrices for climatic factors, species diversity indices, coefficients of variation (CVs, inverse of stabilities), and synchronies (inverse of asynchronies) estimated with all species (**a**) and only
*Appendix 1—figure 3 continued on next page*

*Appendix 1—figure 3 continued*

dominant species (**b**) by considering a three-local-community scenario (similar sampling as in **Figure 1b**, but with three sites in each sample). Significant and marginally significant correlations are marked with \*p<0.05 and #p<0.10, respectively (see 'Materials and methods' for details). Symbols and descriptions can be found in **Box 1** and **Appendix 1—table 1**. Potentially owing to the small sample size (n = 7) of the three-local-community scenario, many significant (or marginally significant) correlations found in the two-local-community scenario (n = 10, **Appendix 1—figure 2**) were nonsignificant for this three-local-community scenario. Thus, we did not further analyze the three-local-community scenario. Dataset, code, and relevant results are available in Figshare at https://doi.org/10.6084/m9.figshare.20281902.

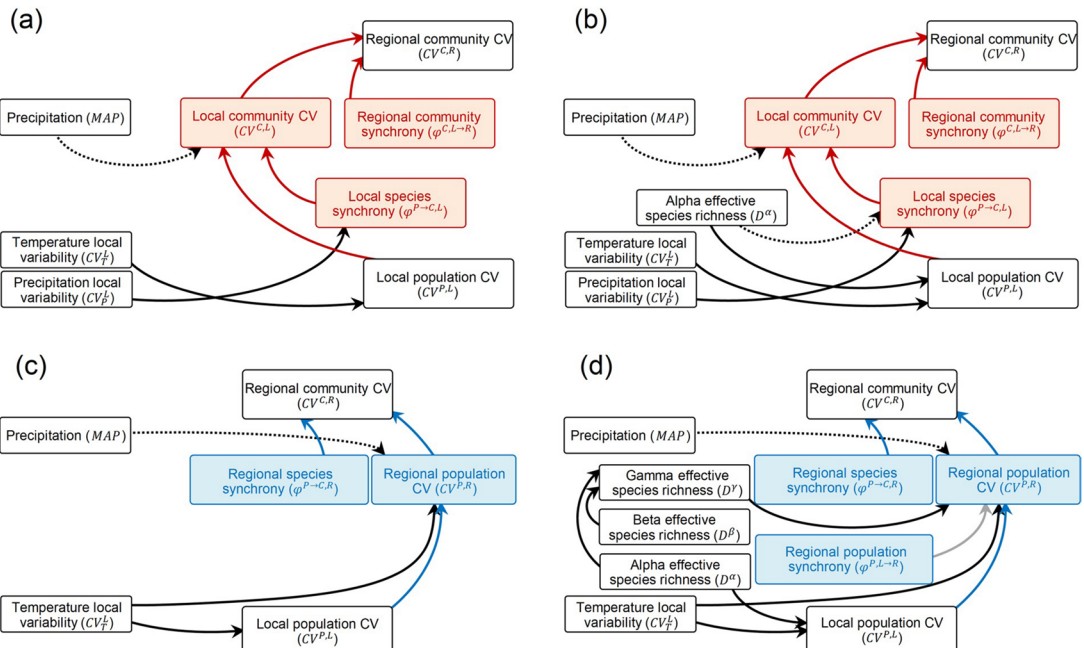

**Appendix 1—figure 4.** Initial path analysis models relating the regional community coefficient of variation (CV, inverse of stability) to its hierarchical components and species diversity indices estimated with all species as well as climatic factors according to the upscaling pathways of aggregating local communities (pathway I, **a, b**) or aggregating regional populations (pathway II, **c, d**). Solid and dashed arrows represent significant (or marginally significant) positive and negative correlation relationships, respectively (**Appendix 1—figure 2a**). Gray solid arrow (regional population CV in relation to regional population synchrony), (**d**) represents nonsignificant positive correlation relationship, which is added in the initial model because it is theoretically proposed (**Wang et al., 2019**). Because (**b**) includes all paths of (**a**) and (**d**) includes all paths of (**c**), only the models shown in (**b**) and (**d**) are further analyzed (details are available in Figshare at https://doi.org/10.6084/m9.figshare.20281902). Symbols and descriptions can be found in **Box 1** and **Appendix 1—table 1**.

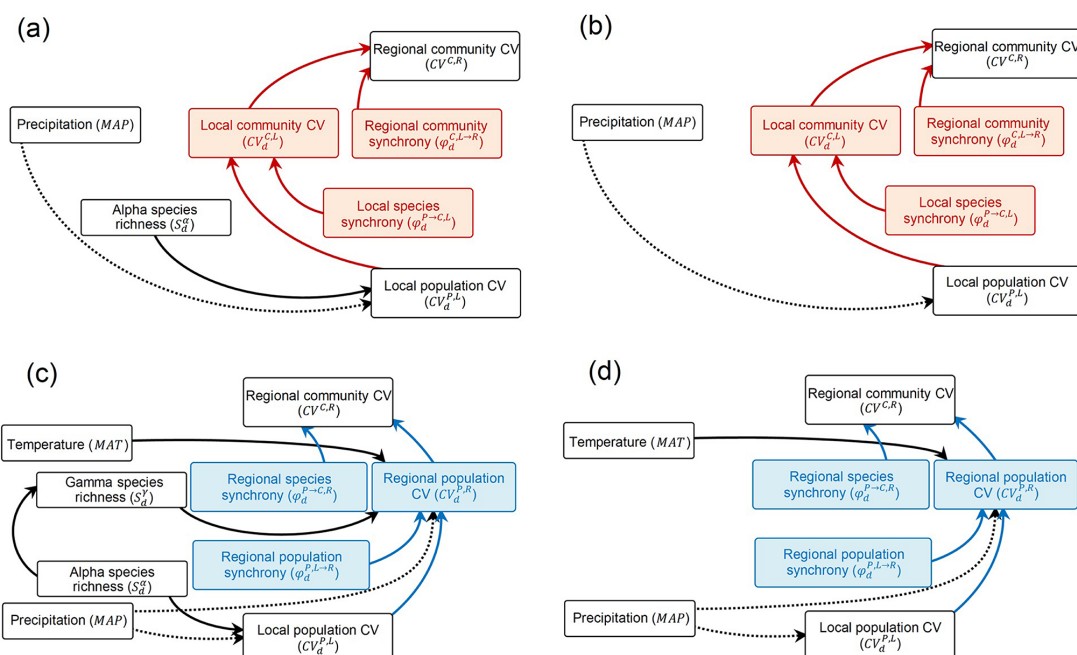

**Appendix 1—figure 5.** Initial path analysis models relating the regional community coefficient of variation (CV, inverse of stability) to its hierarchical components and species diversity indices estimated with only dominant species as well as climatic factors according to the upscaling pathways of aggregating local communities (pathway I, **a, b**) or aggregating regional populations (pathway II, **c, d**). Solid and dashed color arrows represent significant (or marginally significant) positive and negative correlation relationships, respectively (**Appendix 1—figure 2b**). Because (**a**) includes all paths of (**b**) and (**c**) includes all paths of (**d**), only the models shown in (**b**) and (**c**) are further analyzed (details are available in Figshare at https://doi.org/10.6084/m9.figshare.20281902). Symbols and descriptions can be found in **Box 1** and **Appendix 1—table 1**.

