## [Editor Report]

Wang et al. adapt a new statistical framework on a multisite multiyear database to investigate the effects of environmental variables on the temporal stability of plant communities and biomass productivity in Chinese grassland. The authors show that the temporal stability of the region is due to spatial asynchrony of community dynamics. This article will be a landmark in the field as it sets a new methodological framework to study the impacts of global warming in ecosystems and conservation biologists and politicians to design regional policies for land management.

---

## [Decision Letter]

**Decision letter after peer review:**

Thank you for submitting your article "Dominant species determine ecosystem stability across scales in Inner Mongolian grassland" for consideration by *eLife*. Your article has been reviewed by 2 peer reviewers, one of whom is a member of our Board of Reviewing Editors, and the evaluation has been overseen by Detlef Weigel as the Senior Editor. The reviewers have opted to remain anonymous.

We apologize for the delay in having the work reviewed, but it first took a while to recruit reviewers, and then one of the reviewers became unavailable.

*Essential revisions:*

1) Both reviewers agree that the work is very interesting, but also that the framing must be greatly improved. The manuscript is currently too complex and too wordy for readers to fully comprehend its significance.

2) A request from both reviewers is that you revise the usage of the terminology, wording, and analyses. You do not need to reduce any of these but you need to make a better effort at presenting your ideas in a more concise way that appeals to a broad audience, making sure that the readers easily can recognize which of your take-home messages are the most important ones, and which ones are merely secondary ideas.

*Reviewer #1 (Recommendations for the authors):*

Despite my optimism for your work, I provide a few comments that I hope improve the readiness of this manuscript.

1. I understand that a significant part of this manuscript is the implementation (and demonstration) of a new framework and its statistical analyses. And I appreciate short manuscripts, like yours. But the manuscript is rather loaded with terms of increasing complexity. For example, just look at Figure 4! **Dominant species, local scale, population CV!!*. Three-term combinations, with two options per term, make up to 8 comparisons. This increasingly complex terminology makes it difficult to follow, even for people accustomed to reading and writing about time and communities. I found myself bored to death with X x Y comparisons at some point. I thus suggest the authors improve the structure of the manuscript. This can be done either by adding subtitles in each section or by changing the whole structure of the manuscript. For example, work in subsection '1' pathway 1 vs. pathway 2 comparisons with all species. Then work in subsection '2' the same comparison, but now using only dominant species. Similarly, one finds little comfort in the R code. Please mark it as much as possible, especially if your idea is that less versed R people use it in their own datasets.

2. Make a box with ecological level terminology (similar to Table 1, but for ecological terms). For example, synchrony usually has a positive connotation. But in your manuscript, it is negative. You easily understand this negative connotation because you nicely explained it on the first mention. But after so many comparisons and statistical analyses, one gets lost. I am sure that a box with the terminology will help ease this issue.

3. Same with figures. They are OK, but more work can be done to make them more easily understandable. For example, it took me two weeks! to see the link between the colors in the 'population' panels in (b) and the species in (a). I know the journal limits the number of figures and tables, but make a try.

4. When you build your time series, do you have temporal autocorrelation? If so, how do you remove it?

5. The SEM is superb! Congrats.

6. Do you really want to give seven supplement files just for figure 4? This is what I mean when I say you add unnecessary complexity to your manuscript. Maybe add these files to a single general supplementary file. Merge those other 'normal' supplementary files as much as possible.

Line 319. Mention only the 21 sites that you perform analysis upon.

*Reviewer #2 (Recommendations for the authors):*

The authors analyse an impressive dataset of field data collected across Inner Mongolian Grasslands to test theory concerning the mechanisms promoting temporal stability of plant biomass.

Overall, the analyses seem solid, and the paper is based on strong theory, but the overall message is diluted by a large number of different analyses, making the analysis, results, and interpretation confusing in several places.

The unfocused nature of the analysis and presentation of the results makes it difficult to evaluate whether the authors achieve their aims, and whether their results support the conclusions. My general impression is that they do, but the number of different analyses, supplementary results, etc., really complicates the narrative and interpretation.

The paper is an interesting test of theory, and a practical test of the theory outlined in a previous paper (Wang et al.) could be a real asset to anyone aiming to explore the mechanisms promoting temporal stability across scales. The dataset too is a large and potentially useful one.

That said, without a clearer narrative and streamlined set of analyses, it is difficult to interpret the potential impact of this work – which is a shame, because clearly the work put in was considerable. By focusing on only a few key analyses and results, interpretability and potential impact could be much improved.

More specific comments to follow:

Overall, I think the theory is not well enough explained in the main text – it took me several re-reads to understand the different pathways (I and II). I finally understood from supplementary file 2, but could not grasp the concept at all from the main text. So, my suggestion would be to integrate the writing/explanation from supplementary file 2 into the main text in the introduction to more clearly explain the concept (moreover, I think Figure 1 did little to help my understanding, and may in fact have hindered it. I suggest revising/removing Figure 1 to aid clarity).

The analyses too were difficult to follow. I understand that you did several sets of analyses (at different scales and on the full community vs only dominant species etc.), but the justification and explanation of these different analyses are not yet well enough introduced in the introduction or explained in the Methods. Without this, the Results section (and hence interpretation) is currently difficult to follow.

Specifically, when the results for tests of pathways I and II are introduced in the Results section (e.g. L173) it is currently unclear how the authors tested these mechanisms, since the analyses are insufficiently introduced in the introduction (when Methods are presented last, a brief overview of the methods in the final paragraph of the introduction is useful to readers).

One suggestion I have for streamlining your analyses and aiding interpretation is to remove the analyses of climate variables. To me, this feels like a different analysis which aims to answer a different question – indeed, it was not always clear to me what question(s) the authors set out to test. As I read this paper, I was reminded on several occasions of the authors' Proceedings B paper (based on the same dataset) which finds precipitation to be a key driver of stability and also measures synchrony. Perhaps removing the climate analyses here would not only streamline the present paper, but help to distinguish it from the previously published work on the same dataset.

I also wasn't clear on the justification behind constructing large-scale communities from 2 sites (especially when those sites were often far apart)? I understand the aim to calculate β-div from γ/α, but why these communities were summed across only 2 sites rather than all sites was not clear to me. I think better clarification is required in the methods or introduction.

Perhaps the clearest way to improve readability and understanding is to make much more explicit in the introduction, the questions that are being tested and the potential hypotheses for directionality etc. Even then linking those hypotheses to the many figures could help aid interpretability.

---

## [Author Response]

Essential revisions:1) Both reviewers agree that the work is very interesting, but also that the framing must be greatly improved. The manuscript is currently too complex and too wordy for readers to fully comprehend its significance.2) A request from both reviewers is that you revise the usage of the terminology, wording, and analyses. You do not need to reduce any of these but you need to make a better effort at presenting your ideas in a more concise way that appeals to a broad audience, making sure that the readers easily can recognize which of your take-home messages are the most important ones, and which ones are merely secondary ideas.

Thank you very much for your constructive suggestions regarding our manuscript. We understand that our initial presentation was very complex. In the revised manuscript we have tried to reduce all complexity to the minimum needed to convey the main message of how asynchronous dynamics of plant populations and communities can increase regional ecosystem stability. We now use Box 1 to explain the theoretical framework and hypotheses and to provide a glossary of terms. Furthermore, we provide an overview over our analysis approach at the end of Introduction section. We completely rewrote the Results section where we first report results using all species and then results using only dominant species. In the following we explain in detail how we have incorporated all suggestions in the revised manuscript.

Reviewer #1 (Recommendations for the authors):Despite my optimism for your work, I provide a few comments that I hope improve the readiness of this manuscript.1. I understand that a significant part of this manuscript is the implementation (and demonstration) of a new framework and its statistical analyses. And I appreciate short manuscripts, like yours. But the manuscript is rather loaded with terms of increasing complexity. For example, just look at Figure 4! **Dominant species, local scale, population CV!!*. Three-term combinations, with two options per term, make up to 8 comparisons. This increasingly complex terminology makes it difficult to follow, even for people accustomed to reading and writing about time and communities. I found myself bored to death with X x Y comparisons at some point. I thus suggest the authors improve the structure of the manuscript. This can be done either by adding subtitles in each section or by changing the whole structure of the manuscript. For example, work in subsection '1' pathway 1 vs. pathway 2 comparisons with all species. Then work in subsection '2' the same comparison, but now using only dominant species. Similarly, one finds little comfort in the R code. Please mark it as much as possible, especially if your idea is that less versed R people use it in their own datasets.2. Make a box with ecological level terminology (similar to Table 1, but for ecological terms). For example, synchrony usually has a positive connotation. But in your manuscript, it is negative. You easily understand this negative connotation because you nicely explained it on the first mention. But after so many comparisons and statistical analyses, one gets lost. I am sure that a box with the terminology will help ease this issue.

Thank you very much for these comments and suggestions. We have simplified our terminology and made sure only one term is consistently used for the same thing. We added a box explaining the theoretical framework and hypotheses and providing a glossary of the terms used in the text. In addition, we added a paragraph at the end of the Introduction section to provide an overview of our analysis approach.

According to your suggestions, we separated the Results section into two parts. One part focuses on the analysis using all species and the other on the analysis using only dominant species.

In addition, we annotated the R code with comments and added separations. To ease the read of our manuscript and make reader focus on the main findings, details about structural equation models (SEMs) and the stepwise elimination of non-significant paths were removed from the manuscript, but delivered as a summary file to a third-party data deposition (Figureshare at https://doi.org/10.6084/m9.figshare.20281902). Based on the same consideration, SEMs and GLMs used to further examining species diversity impacts were also removed, but, together with our dataset and R code, delivered as a summary file for data deposition.

3. Same with figures. They are OK, but more work can be done to make them more easily understandable. For example, it took me two weeks! to see the link between the colors in the 'population' panels in (b) and the species in (a). I know the journal limits the number of figures and tables, but make a try.

We removed Figure 1 of the previous version. In this revised version, we added a diagram in Box 1 (accompanying figure of Box 1) to explain the theoretical framework as well as theoretically proposed impacts of species diversity. In addition, we also simplified colors used in diagrams (such as Figure 3 and Supplementary file 2–Figure 3–4 of this revised version). In the revised diagrams, different upscaling pathways are colored with red for pathway I and blue for pathway II.

4. When you build your time series, do you have temporal autocorrelation? If so, how do you remove it?

We conducted autocorrelation analyses and found no autocorrelations for each surveyed site. Please see Author response image 1.

**Author response image 1. sa2fig1:** Results of autocorrelation analyses for community productivity at each surveyed site. Dashed lines indicate 95% confidence bands. There were no significant autocorrelations at any site.

5. The SEM is superb! Congrats.6. Do you really want to give seven supplement files just for figure 4? This is what I mean when I say you add unnecessary complexity to your manuscript. Maybe add these files to a single general supplementary file. Merge those other 'normal' supplementary files as much as possible.

We simplified supplementary information for these diagrams. In this revised version, only correlation matrices and initial path analysis models constructed based on these correlation analyses are provided as supplements (Supplementary file 2).

Line 319. Mention only the 21 sites that you perform analysis upon.

Thank you. We revised it as suggested.

Reviewer #2 (Recommendations for the authors):The authors analyse an impressive dataset of field data collected across Inner Mongolian Grasslands to test theory concerning the mechanisms promoting temporal stability of plant biomass.Overall, the analyses seem solid, and the paper is based on strong theory, but the overall message is diluted by a large number of different analyses, making the analysis, results, and interpretation confusing in several places.The unfocused nature of the analysis and presentation of the results makes it difficult to evaluate whether the authors achieve their aims, and whether their results support the conclusions. My general impression is that they do, but the number of different analyses, supplementary results, etc., really complicates the narrative and interpretation.The paper is an interesting test of theory, and a practical test of the theory outlined in a previous paper (Wang et al.) could be a real asset to anyone aiming to explore the mechanisms promoting temporal stability across scales. The dataset too is a large and potentially useful one.That said, without a clearer narrative and streamlined set of analyses, it is difficult to interpret the potential impact of this work – which is a shame, because clearly the work put in was considerable. By focusing on only a few key analyses and results, interpretability and potential impact could be much improved.

Thank you very much for your constructive suggestions. We have revised the paper throughout to increase the focus and readability. To help readers to easily understand stability theory and analysis as used in this study, we added Box 1 which combines the theoretical framework with hypotheses, especially about proposed effects of species diversity on stability, and provides a glossary of terms. In addition, we summarize our approach at the end of the Introduction section and in the Results section first present the analysis using all species and then the analysis using only dominant species. Furthermore, to focus on our main findings, we removed detailed analyses (but deliver them as summary files together with dataset and R script to a third-party data deposition). Finally, we added calculations of CV and synchrony across spatial scales to the Methods section.

More specific comments to follow:Overall, I think the theory is not well enough explained in the main text – it took me several re-reads to understand the different pathways (I and II). I finally understood from supplementary file 2, but could not grasp the concept at all from the main text. So, my suggestion would be to integrate the writing/explanation from supplementary file 2 into the main text in the introduction to more clearly explain the concept (moreover, I think Figure 1 did little to help my understanding, and may in fact have hindered it. I suggest revising/removing Figure 1 to aid clarity).The analyses too were difficult to follow. I understand that you did several sets of analyses (at different scales and on the full community vs only dominant species etc.), but the justification and explanation of these different analyses are not yet well enough introduced in the introduction or explained in the Methods. Without this, the Results section (and hence interpretation) is currently difficult to follow.Specifically, when the results for tests of pathways I and II are introduced in the Results section (e.g. L173) it is currently unclear how the authors tested these mechanisms, since the analyses are insufficiently introduced in the introduction (when Methods are presented last, a brief overview of the methods in the final paragraph of the introduction is useful to readers).

In this revised version, we removed the previous Figure 1 but instead added Box 1 with an accompanying figure to show the theoretical framework relating regional community CV to its hierarchical components along the two alternative pathways I or II. In addition, this Box also provides a glossary for terminology.

According to your suggestions, we added a paragraph at the end of the Introduction section (lines 145–155) to provide an overview over our analysis procedure in the current study. Briefly, we investigated how asynchronous population dynamics among species, especially dominant species, contributed to the stability of local and regional communities. We also tested how local and regional community dynamics were driven by species diversity or affected by climatic factors, such as precipitation as well as its temporal variation. These examinations were based on path-analytic diagrams constructed according to theoretical knowledge (shown in Box 1 and its accompanying figure).

Based on the new Introduction section, we further revised the Results section. Specifically, the new Results section has two parts. The first part focuses on the analysis using all species and the second on the analysis using only dominant species. This should be helpful to easily follow examinations conducted in the current study.

One suggestion I have for streamlining your analyses and aiding interpretation is to remove the analyses of climate variables. To me, this feels like a different analysis which aims to answer a different question – indeed, it was not always clear to me what question(s) the authors set out to test. As I read this paper, I was reminded on several occasions of the authors' Proceedings B paper (based on the same dataset) which finds precipitation to be a key driver of stability and also measures synchrony. Perhaps removing the climate analyses here would not only streamline the present paper, but help to distinguish it from the previously published work on the same dataset.

Thank you for this suggestion. However, we would like to keep precipitation as an environmental covariate in the analysis because it has such a strong effect on synchronizing plant dynamics in the arid study region. In our previous publication, we found destabilizing effects of decreasing mean and increasing variability in precipitation at the local scale. This had motivated us to perform the current study to further explore whether these mechanisms of precipitation scale up to the regional scale and whether novel mechanisms arise, such as whether regional synchronous dynamics of precipitation regulate regional population synchrony or regional community synchrony. To the best of our knowledge, this has not been investigated previously. By combining our large-scale dataset and a spatial stability framework, our current study confirmed that the local-scale mechanisms of precipitation do indeed scale up to regional scale. We tried to clarify the overall analysis approach further, so that the influence of precipitation as environmental covariate can be more easily compared with the influence of stability drivers that are intrinsic to the plant community such as species diversity. We believe precipitation variability is also relevant in the context of management decisions, because some models predict increased precipitation variability in the future.

I also wasn't clear on the justification behind constructing large-scale communities from 2 sites (especially when those sites were often far apart)? I understand the aim to calculate β-div from γ/α, but why these communities were summed across only 2 sites rather than all sites was not clear to me. I think better clarification is required in the methods or introduction.

We considered this issue by adding correlation analyses based on a 3-site scenario in Supplementary file 2–Figure 2. However, we found that the statistical power of this analysis was low, because with 21 sites a maximum of only 7 non-overlapping triplets of local communities could be assembled. Therefore, we did not further analyze the 3-site scenario as well as scenarios with more than 3 sites within a regional community.

Perhaps the clearest way to improve readability and understanding is to make much more explicit in the introduction, the questions that are being tested and the potential hypotheses for directionality etc. Even then linking those hypotheses to the many figures could help aid interpretability.

In this revised version, we added Box 1 with an accompanying figure to explain the theoretical framework and hypotheses underpinning our work. In addition, we also added a paragraph at the end of the Introduction section to introduce our analyses (lines 145–155).